



**Trends analysis of PM source contributions and chemical tracers in NE Spain during 2004 - 2014: A multi-exponential approach.**
*Marco Pandolfi [1,*], Andrés Alastuey [1], Noemi Pérez [1], Cristina Reche [1], Iria Castro [1], Victor Shatalov [2] and Xavier Querol[1]*
*[1] Institute of Environmental Assessment and Water Research, c/ Jordi-Girona 18-26, 08034 Barcelona, Spain*
*[2] Meteorological Synthesizing Centre – East, 2nd Roshchinsky proezd, 8/5, 115419 Moscow, Russia*
*Corresponding author: marco.pandolfi@idaea.csic.es*
**Abstract**
In this work for the first time data from two twin stations (Barcelona, urban background, and Montseny,
regional background), located in NE of Spain, were used to study the trends of the concentrations of different
chemical species in $PM_{10}$ and $PM_{2.5}$ along with the trends of the $PM_{10}$ source contributions from Positive Matrix
Factorization (PMF) model. Eleven years of chemical data (2004–2014) were used for this study. Trends of both
specie concentrations and source contributions were studied using the Mann-Kendall test for linear trends and
a new approach based on multi-exponential fit of the data. Despite the fact that different PM fractions ($PM_1$,
$PM_{2.5}$, $PM_{10}$) showed linear decreasing trends at both stations, the contributions of specific sources of pollutants
and the related chemical tracers showed exponential (single or double) decreasing trends. The different types
of trends observed reflected the different effectiveness and/or time of implementation of the measures taken
to reduce the concentrations of atmospheric pollutants (i.e. those implemented in the Industrial Emission
Directives and in the Large Combustion Plants Directive). Moreover, the trends of the contributions from
specific sources such as those related with industrial activities and with primary energy consumption mirrored
the effect of the financial crisis in Spain from 2008. The sources that showed statistically significant downward
trends at both Barcelona (BCN) and Montseny (MSY) during 2004-2014 were *Ammonium sulfate*, *Ammonium
nitrate*, and *V-Ni bearing* source. The contributions from these sources decreased exponentially during the
considered period indicating that the observed decrease was not gradual and consistent over time. Moreover,
statistically significant decreasing trends were observed for the contributions to PM from the *Industrial/Traffic*
source at MSY (mixed metallurgy and road traffic) and from the *Industrial* (metallurgy mainly) source at BCN.
These sources were clearly linked with anthropogenic activities and the observed decreasing trends confirmed
the effectiveness of pollution control measures implemented at EU or regional/local levels. The general trends
observed for the calculated PMF source contributions well reflected the trends observed for the chemical
tracers of these pollutant sources.



## 1. Introduction

Meeting the air quality (AQ) standards is one of the major environmental objectives to protect people from breathing air with high levels of pollution. Many studies have been published in these last years showing clearly that the concentrations of particulate matter (PM), and other air pollutants such as sulphur dioxide ($SO_2$) and carbon monoxide (CO), have markedly decreased during the last 15 years in many European Countries (EEA, 2013; Barmpadimos et al., 2012; Cusack et al., 2012; Querol et al., 2014; Guerreiro et al., 2014 among others). Cusack et al. (2012) reported the reduction in $PM_{2.5}$ concentrations observed at regional background (RB) stations in Spain and across Europe, and, in most cases, the observed reduction was gradual and consistent over time, implying the success of cleaner anthropogenic activities. Barmpadimos et al. (2012) have also shown that $PM_{10}$ concentrations decreased at a number of urban background (UB) and rural background stations in five European countries. Henschel et al. (2013) reported the dramatic decrease in $SO_2$ levels across six European cities, reflecting the reduction in sulphur content in fuels, as part of EU legislation, coupled with the shift towards the use of cleaner fuels. EEA (2013) also reported general decreases in $NO_2$ concentrations even if lower compared to PM. However, Henschel et al. (2015) showed that the $NO_x$ concentrations at traffic sites in many EU cities remained unchanged underlining the need of further regulative measures to meet the air quality standards for this pollutant. In fact an important proportion of the European population lives in areas exceeding the AQ standards for the annual limit value of $NO_2$, the daily limit value of $PM_{10}$ and the health protection objective of $O_3$ (EEA, 2013; 2015). $PM_{10}$ and $NO_2$ are still exceeded mostly in urban areas, and especially at traffic sites (Harrison et al., 2008; Williams and Carslaw, 2011; EEA, 2013; among others). In Spain for example it has been reported that more than 90% of the $NO_2$ exceedances are attributed to road traffic emissions (Querol et al., 2012). Guerreiro et al. (2014) furthermore evidenced notable reduction of ambient air concentration of $SO_2$, CO and Pb using data available in Airbase (EEA, 2013) and covering 38 European countries. Querol et al. (2014) reported trends for 73 measurement sites across Spain including RB, UB, traffic stations (TS) and Industrial sites (IND). They observed marked downward concentration trends for $PM_{10}$, $PM_{2.5}$, CO and $SO_2$ at most of the RB, UB, TR and IND sites considered. Similarly, Salvador et al. (2012) detected a statistically significant downward trends in the concentrations of $SO_2$, $NO_x$, CO and $PM_{2.5}$ at most of the urban and urban-background monitoring sites in the Madrid metropolitan area during 1999-2008. Cusack et al. (2012) and Querol et al. (2014) have also shown the highly statistically significant decreasing trends observed at regional level in NE Spain for many trace elements since 2002 (Pb, Cu, Zn, Mn, Cd, As, Sn, V, Ni, Cr).

The observed reduction of air pollutants across Europe is the results of efficient emission abatement strategies as for example those implemented in the Industrial Emission Directives (IPPC Integrated Pollution Prevention and Control and subsequent Industrial Emission Directives 1996/61/EC and 2008/1/EC), the Large Combustion Plants Directive (LCPD; 2001/80/EC), the EURO standards on road traffic emission (1998/69/EC, 2002/80/EC, 2007/715/EC), the IMO (International Maritime Organization) directive on sulphur content in fuel, and $SO_x$ and $NO_x$ emissions from ships (IMO, 2011; Directive 2005/33/EC). Additionally, the financial crisis, causing mainly a



reduction of the primary energy consumption from 2008-2009, contributed to the decrease of the ambient
concentration of pollutants observed in Spain (Querol et al., 2014).
Moreover, national and regional measures for AQ have been taken in many European Countries. In Spain a
national AQ plan was approved in 2011 and updated in 2013 by the Council of Ministers of the Government of
Spain. Furthermore, 45 regional and 3 local (city scale) AQ plans have been implemented since 2004 in Spain.
These AQ Plans mostly focused on improving AQ at major city centers or specific industrial areas.
For the aforementioned reasons, it is especially attractive the feasibility of studying the trends of the
contributions to PM mass from specific pollutant sources along with the trends of the chemical tracers of these
sources.
A repository of previous works studying the trends of atmospheric pollutants is reported in the Wikipedia of *The*
*Task Force on Measurements and Modelling* (TFMM; [https://wiki.met.no/emep/emep-](https://wiki.met.no/emep/emep-)
[experts/tfmmtrendpublis](experts/tfmmtrendpublis)). The TFMM together with the *Task Force on Emission Inventories and Projections*
(TFEIP), the *Task Force on Integrated Assessment Modelling* (TFIAM), and *Task Force on Hemispheric Transport*
*of Air Pollution* (TFHTAP) provide a fora for discussion and scientific exchange in support of the EMEP (*European*
*Monitoring and Evaluation Programme*; http://www.emep.int/) work plan which is a scientifically based and
policy driven programme under the *Convention on Long-range Transboundary Air Pollution* (CLRTAP;
http://www.unece.org/env/lrtap/lrtap_h1.html) promoting the international co-operation to solve
transboundary air pollution problems. The TFMM was established in 2000 to evaluate measurements and
modeling and to further develop working methods and tools. In this contest, five EMEP Centers are undertaking
efforts in support of the EMEP work plan, namely the *Centre on Emission Inventories and Projections* (CEIP;
http://www.ceip.at/), the *Chemical Coordinating Centre* (CCC; http://www.nilu.no/projects/ccc/), the
*Meteorological Synthesizing Centre – West* (MSC-W; http://emep.int/mscw/index_mscw.html), the
*Meteorological Synthesizing Centre – East* (MSC-E; http://www.msceast.org/), and the *Centre for Integrated*
*Assessment Modelling* (CIAM; http://www.iiasa.ac.at/~rains/ciam.html). In 2014, the TFMM initiated a
dedicated exercise to assess the efficiency of air pollution mitigation strategies over the past 20 years to assess
the benefit of the CLRTAP main policy instrument. Within this exercise a software was made available by
EMEP/MSC-E Center aiming at studying non-linear trends and specifically using multi-exponential fits
([https://wiki.met.no/emep/emep-experts/tfmmtrendmethods](https://wiki.met.no/emep/emep-experts/tfmmtrendmethods)).
For what we are concerned in the majority of studies dealing with trend analysis, linear fits were applied for
example by using Mann-Kendall or Theil-Sen methods (Theil, 1950; Sen, 1968), the latter being available for
example in the Openair software (Carslaw, 2012; Carslaw and Ropkins, 2012). However, linear fit of data does
not always properly represent the observed trends. As we will show, different abatement strategies and periods
of implementation may change from one pollutant to another thus leading to different trends for different
pollutants, even over the same period. Thus, non-linear fit of the data may be at times strongly recommended.





In this work we studied the trends of source contributions to $PM_{10}$ and specific chemical species in both $PM_{10}$
and $PM_{2.5}$ using both the consensus methodology (Mann-Kendall) and the multi-exponential approach. PM
chemical speciated data collected from 2004 to 2014 at regional (Montseny; NE Spain) and urban (Barcelona,
NE Spain) sites were used with this aim. The selected period allowed for trend analysis at these twin stations
over a common period. The Positive Matrix Factorization (PMF) model was used to apportion ambient $PM_{10}$
concentrations into pollutant sources. The PMF model, as other Receptor Models (RM), is widely used being a
powerful tool to help policy makers to design more targeted approaches to protecting public health. Thus, the
novelty of this study lies mainly in a) the opportunity to study the trends of source contributions from PMF
model at two twin stations representative of the urban and regional environments in the Western
Mediterranean, and, b) in the use of a novel non-linear approach for trend studies.

**2. Measurement sites and Methodology**
**2.1 Measurement sites**
The Montseny measurement station (MSY, 41°46'45.63" N, 02°21'28.92" E, 720 m a.s.l.) is a regional
background site in NE of Spain (Figure 1). The MSY station is located within a regional natural park about 50 km
to the NNE of the city of Barcelona (BCN) and 25 km from the Mediterranean coast. This site is representative of
the typical regional background conditions of the Western Mediterranean Basin (WMB) characterized by severe
pollution episodes affecting not only the coastal sites closest to the emission sources, but also the more
elevated rural and remote areas land inwards due to thermally driven winds (i.e. Pérez et al., 2008; Pey et al.,
2010; Pandolfi et al., 2011; 2014). This station is part of ACTRIS (www.actris.net) and GAW (www.wmo.int/gaw)
networks, EMEP (http://www.emep.int/) and the measuring network of the Government of Catalonia.
The Barcelona measurement station (BCN, 41°23'24.01" N, 02°06'58.06" E, 68 m a.s.l.) is an urban background
measurement site influenced by vehicular emissions from one of the main avenues of the city (Diagonal
Avenue) located at a distance of around 300 m (cf. Fig. 1). The BCN measurement site is part of the Air Quality
measuring network of the Government of Catalonia. The Metropolitan Area of Barcelona (BMA), with nearly 4.5
million inhabitants, covers an 8 km wide strip between the Mediterranean Sea and the coastal mountain range.
Several industrial zones, power plants, and highways are located in the area, making this region to one of the
most polluted in the WMB (i.e. Querol et al., 2008; Amato et al., 2009; Pandolfi et al., 2012; 2013; 2014). At BCN
the location of the measuring station changed in 2009 when it was moved by around 500 m (cf. Fig. 1). The
effect of this change on PM measurements performed at BCN will discuss later.







**2.2 Real-time and gravimetric PM measurements**
Real-time PM concentrations were continuously measured at 1h resolution by optical particle counters (OPC)
using GRIMM spectrometers (GRIMM 180 at MSY, and GRIMM 1107, 1129 and 180 at BCN). Hourly PM
concentrations were corrected by comparison with 24h gravimetric mass measurements of $PM_x$ (Alastuey et al.,

141     2011).

For gravimetric measurements 24h PMx samples were collected at both stations every 3-4 days on 150 mm
quartz micro-fiber filters (Pallflex QAT and Whatman) with a high-volume (Hi-Vol) samplers (DIGITEL DH80
and/or MCV CAV-A/MSb at 30 $m^3h^{-1}$). The mass of $PM_{10}$ and $PM_{2.5}$ samples collected on filters was determined
using the EN 12341 and the EN14907 gravimetric procedures, respectively.

**2.2.1 PM chemical speciated data**
Once the gravimetric mass was determined from filters, the samples were analyzed with different techniques
including acidic digestion (½ of each filter; $HNO_3$:HF:$HClO_4$), water extraction of soluble anions (¼ of each filter),
and thermal-optical analysis (1.5 $cm^2$ sections). Inductively Coupled Atomic Emission Spectrometry, ICP-AES,
(IRIS Advantage TJA Solutions, THERMO) was used for the determination of the major elements (Al, Ca, Fe, K,
Na, Mg, S, Ti, P), and Inductively Coupled Plasma Mass Spectrometry, ICP-MS, (X Series II, THERMO) for the
trace elements (Li, Ti, V, Cr, Mn, Co, Ni, Cu, Zn, As, Se, Rb, Sr, Cd, Sn, Sb, Ba, rare earths, Pb, Bi, Th, U). Ionic
Chromatography was used for the concentrations of $NO_3^-$, $SO_4^{2-}$ and $Cl^-$, whereas $NH_4^+$ was determined using a
specific electrode MODEL 710 A+, THERMO Orion. The levels of OC and EC were determined by a thermal-
optical carbon analyzer (SUNSET), using protocol EUSAAR_2. Other analytical details may be found in Querol et
al. (2009).
Following the above procedures, $PM_{10}$ and $PM_{2.5}$ chemical speciated data were obtained at MSY for the period
2004-2014 resulting in 1093 and 794 samples, respectively. At BCN $PM_{10}$ and $PM_{2.5}$ data were obtained during
2004-2014 resulting in 1037 and 1063 samples, respectively.

**2.3 Positive Matrix Factorization (PMF) model.**
The PMF model (PMFv5.0, EPA) was used on the collected daily speciated data for source identification and
apportionment in $PM_{10}$ at both sites. Detailed information about the PMF model can be found in literature
(Paatero and Tapper 1994; Paatero 1997; Paatero and Hopke 2003; Paatero et al. 2005). The PMF model is a
factor analytical tool reducing the dimension of the input matrix in a limited number of factors (or sources) and
it is based on the weighted least-squares method. Thus, most important in PMF applications is the estimation of
uncertainties of the chemical species included in the input matrix. In the present study, individual uncertainties



and detection limits were calculated as in Escrig et al. (2009) and Amato et al. (2009). Thus, both the analytical
uncertainties and the standard deviations of species concentrations in the blank filters were considered in the
uncertainties calculations. The signal-to-noise ratio (S/N) was estimated starting from the calculated
uncertainties and used as a criteria (S/N >2) for selecting the species used within the PMF model. In order to
avoid any bias in the PMF results, the data matrix was uncensored (End user's guide to multilinear engine
applications from Pentti Paatero). The PMF was run in robust mode (Paatero 1997), and rotational ambiguity
was handled by means of the $F_{PEAK}$ parameter (Paatero et al. 2005). The optimal number of sources was selected
by inspecting the variation of the objective function $Q$ (defined as the ratio between residuals and errors in each
data value) with varying number of sources (i.e. Paatero et al., 2002) and by studying the physical
meaningfulness of the calculated factors.
**2.4 Mann-Kendall (MK) fit**
The purpose of the Mann-Kendall (MK) test (Mann 1945, Kendall 1975, Gilbert 1987) is to statistically assess if
there is a monotonic upward or downward trend of the variable of interest over time. A monotonic upward
(downward) trend means that the variable consistently increases (decreases) through time.

**2.5 Multi-exponential (ME) fit**
A Program aiming at studying trends of time series of air pollution in the multi-exponential form was developed
within the TFMM by MSC-E group (Shatalov et al., 2015). Annual, monthly and daily resolution data can be
analyzed with the help of this program. Since in this paper we will apply the program to annual averages of
specie concentrations and source contributions, we restrict the description of the multi-exponential
approximations for this case. In particular, seasonal variations are not included into consideration. The basic
equations solved by the program for this particular case (annual averages) are reported below:

$$C_t = a_1 \cdot \exp\left(-\frac{t}{\tau_1}\right) + a_2 \cdot \exp\left(-\frac{t}{\tau_2}\right) + \ldots + a_n \cdot \exp\left(-\frac{t}{\tau_n}\right) + \omega_t \quad (1)$$

Where, $C_t$ are the values of the considered time series, with $t$ = 1, …, N, N being the length of the series (years),
$\tau_n$ are the characteristic times of the considered exponential, and $a_n$ are constants. In the case of single
exponential decay (n=1) the characteristic time $\tau$ is the time at which the pollutant concentration is reduced to
1/e (= 0.3678) times its initial value. Both $\tau_n$ and $a_n$ are calculated by the program by means of the least square
method minimizing the residue $\omega$. The number of exponential terms that should be included into the
approximation can be evaluated using F-statistics (i.e. Smith, 2002). For example, the $F$-statistics for the
evaluation of the statistical significance of the second term in equation (1) for n = 2 can be calculated as:




$F = \frac{(SS_1 - SS_2)}{2 \cdot s}$       (2)

where $SS_1$ and $SS_2$ are sums of squares of residual component for approximations with one and two exponential
terms, respectively, and $s$ is the estimate of standard deviation of residual component. This statistics follows
approximately the Fisher distribution with 2 and $N - 2$ degrees of freedom. Second exponential is considered to
be significant if $F$ exceeds the corresponding threshold value at the chosen significance level.
The following parameters can be calculated from equation (1):
-    Total Reduction (TR):      $TR = \frac{(C_{beg} - C_{end})}{C_{beg}} = 1 - \frac{C_{end}}{C_{beg}}$       (3)
-    Annual reduction for year $i$:   $R_i = \frac{\Delta C_i}{C_i} = 1 - \frac{C_{i+1}}{C_i}$       (4)
-    Average annual reduction:    $R_{av} = 1 - \left(\frac{C_{end}}{C_{beg}}\right)^{\frac{1}{N-1}}$       (5)
The formula for calculation of average annual reduction takes into account that the ratio $C_i+1$ / $C_i$ is a
multiplicative quantity, so that geometrical mean of ratios should be used.
The MSC-E also proposed a statistic which can be used to check if the trends are linear or not (Non-Linearity:
*NL*). More detailed description of the multi-exponential approach is available in the TFMM wiki and in the MSC-
E Technical report 2015 (Shatalov et al., 2015).

**3. Results**
Results will be presented and discussed in the following order: First (*Paragraph 3.1*), we will compare the trends
at both stations of $PM_x$ concentrations from optical counters (OPC; annual data coverage around 90%) and from
24h gravimetric samples (filters; annual data coverage around 20-30%). This comparison will demonstrate the
feasibility of studying trends of pollutant concentrations from filters analyses despite the relatively low annual
data coverage. Second (*Paragraph 3.2*), we will compare the magnitude of the trend of $PM_{2.5}$ concentrations at
MSY during 2004-2014 (period selected for this study) with the magnitude of trends calculated at the same
station over different periods, namely 2002-2010 (the period used in Cusack et al., 2012) and 2002-2014
(representing the largest period of gravimetric $PM_{2.5}$ measurements available so far at MSY station). This
comparison was performed in order to study the effects of meteorology on the magnitude of the trends over
relatively short periods. The gravimetric concentrations of $PM_{2.5}$ were used with this aim. Third, we will present
and discuss the trends at both stations of chemical species in both $PM_{10}$ and $PM_{2.5}$ from 24h filter analyses
(*Paragraph 3.3*). Fourth, we will discuss the sources of pollutants identified by PMF model in $PM_{10}$ at both sites



(*Paragraph 4.0*). Finally, we will present and discuss the trends of $PM_{10}$ source contributions at BCN and MSY
(*Paragraph 4.1*). In *Paragraph 4.1* we will provide possible explanations for the observed trends.

**3.1 Trends of PM: Comparison between gravimetric and real-time optical measurements**

In this section we evaluate the feasibility of studying the trends of PM concentrations from gravimetric samples
by comparing the trends of 24h PM concentrations from filters to those obtained using daily averaged real-time
data from optical counters (OPC). In fact, the annual coverage of gravimetric measurements was around 30% at
both sites, whereas the OPC measurements had an annual coverage of around 90%. Note that the
recommended annual data coverage for trend studies is typically 75%.
Table 1 reports the trends (during 2004-2014) of annual mean concentrations of different PM size fractions
(from both OPC and gravimetric measurements) at both BCN and MSY calculated using both the Mann-Kendall
(MK) and multi-exponential (ME) fits. Gravimetric $PM_1$ concentrations were not available for the whole period
and only OPC data were reported in Table 1. Similarly, calculated $PM_x$ concentrations ($PM_{1-10}$ and $PM_{2.5-10}$) and
PM ratios ($PM_1/PM_{10}$ and $PM_{2.5}/PM_{10}$) were reported only for OPC data. Following the MK test, statistically
significant decreasing trends were observed for all PM size fractions at BCN (ranging from        -2.18 %/yr with
p<0.05 for $PM_{2.5-10}$ to -3.89 %/yr with p<0.001 for $PM_{2.5}$), whereas at Montseny only the $PM_{2.5}$ fraction showed a
statistical significant decreasing trend (-1.87 %/yr, p<0.1 from gravimetric measurements). The higher
statistically significance of the $PM_x$ trends observed at BCN compared to MSY was likely due to the change of
the measuring station in 2009 in BCN (cf. Fig. 1). Based on the comparison between simultaneous $PM_x$ chemical
speciated data collected at both BCN measurement sites during 1 month (not shown) we concluded that after
2009 the BCN measuring site was less affected by mineral matter and, to a lesser extent, by road traffic
emissions both being important sources of PM in Barcelona. In Figure 1 we highlighted the proximity of the BCN
measuring station before 2009 to an unpaved parking and different construction works. The effect of the
change of the station in BCN in 2009 on $PM_{10}$ gravimetric measurements was reported in Supporting
Information (Figure SI-1). Thus, we will discuss here the magnitude of the trends of $PM_x$ concentrations at MSY
only. However, despite the change of station, the comparison between BCN and MSY for specific chemical
species and pollutant sources not linked with these two pollutant sources (mineral matter and road traffic
emissions) was possible.
Table 1 shows that the trends of different $PM_x$ size fractions at MSY were linear (L) during the period 2004-
2014. In fact the non-linearity parameter (NL), which is an estimation of how much the trend differs from the
linear one, was always lower than 10% which was used as threshold for non-linearity (Shatalov et al., 2015). The
characteristic times ($\tau$) from ME fit of $PM_x$ reported in Table 1 were not directly comparable between BCN and
MSY again because of the change of the station occurred in BCN in 2009. As a consequence the characteristic
times ($\tau$) for $PM_1$, $PM_{2.5}$ and $PM_{10}$ at BCN were much lower than at MSY indicating a steeper decrease of PMx





concentrations at BCN compared to MSY (cf. Eq. 1 and Figure SI-1 in Supporting Information). Moreover, much
higher NL and total reductions (TR) were obtained at BCN compared to MSY.
Following the MK test, at MSY station the $PM_1$ fraction from OPC showed non-statistically significant decreasing
trend (-1.25 %/yr), whereas for the $PM_{10}$ fraction (OPC) no trend was observed (0.0 %/yr). The magnitude of the
trends of $PM_{10}$ and $PM_{2.5}$ concentrations at MSY from gravimetric measurements (-0.47 %/yr and -1.87 %/yr,
respectively) confirmed what observed using data from OPC (0.00 %/yr and -1.78 %/yr, respectively).
Similarities were also observed between OPC and gravimetric measurements of $PM_{2.5}$ and $PM_{10}$ at BCN (cf.
Table 1). Thus, despite the different data coverage the magnitudes of the trends calculated from OPC and
gravimetric measurements were very similar. We will show later (*Paragraph 4.1*) that despite the fact that the
gravimetric concentrations of $PM_{10}$ at MSY only slightly decreased monotonically with time (-0.47 %/yr with
p>0.1; cf. Table 1), the contributions from specific $PM_{10}$ pollutant sources from PMF model related with
anthropogenic activities showed non-linear (i.e. exponential) statistically significant decreasing trends. For
example the *Ammonium sulfate* source contribution to $PM_{10}$ at MSY decreased at the rate of -2.02 %/yr with
p<0.05 from MK test and double exponential fit of the data was needed; cf. Table 5). At MSY the $PM_{1-10}$ and
$PM_{2.5-10}$ size fractions showed non-statistically significant increasing trends of around 0.9 %/yr and 0.62 %/yr,
respectively. The decreasing trends of these two PM size fractions at BCN (-2.34 %/yr and -2.18 %/yr for $PM_{1-10}$
and $PM_{2.5-10}$, respectively) were statistically significant however, as already noted, these decreases were largely
explained by the change of the BCN measuring station in 2009. Moreover, at MSY non significant decreasing
trends were observed for the ratio $PM_1/PM_{10}$, with magnitude of -1.25 %/yr and for the ratio $PM_{2.5/10}$ decreasing
at the rate of -0.62 %/yr. At MSY, on average, the residual component (RC) was quite low and around 8-26% for
all PM fractions. Total reductions at MSY (TR) ranged from 8-12% for $PM_{10}$ (not statistically significant) to 24%
for $PM_{2.5}$ (statistically significant at p<0.1).

**3.2 Trends of PM: Comparison among different periods**
In this section we compared the magnitude of the trends of gravimetric $PM_{2.5}$ concentrations at MSY over
different periods (2002-2010; 2004-2014 and 2002-2014) in order to evaluate the effects of meteorology-driven
interannual variability on PM concentrations and trends. Note that the period 2004-2014 was the period chosen
for trends analysis in this study given that gravimetric $PM_{2.5}$ measurements at BCN were available since 2004.
Conversely, at MSY $PM_{2.5}$ gravimetric measurements started in 2002. Figure 2 shows the trends of $PM_{2.5}$
concentrations at MSY calculated using both MK (Fig. 2a) and ME (Fig. 2b) fits for the three different periods.
The period 2002-2010 was the period considered in the paper from Cusack et al. (2012) presenting the trends of
$PM_{2.5}$ gravimetric mass and chemical species in $PM_{2.5}$ at MSY. The period 2002-2014 is the largest period
available so far at MSY with $PM_{2.5}$ filter measurements. The trend observed at MSY for the $PM_{2.5}$ fraction during
2004-2014 confirmed what already observed by Cusack et al. (2012) at the same station for the period 2002 –
2010. In Cusack et al. (2012) the MK test provided a decreasing trend of around -3.4 %/yr at 0.01 significance



level during 2002 – 2010. Considering the period 2004 – 2014, a decreasing trend of -1.87 %/yr at 0.1
significance level was observed. Thus, a statistically significant trend for $PM_{2.5}$ at regional level can be confirmed
even considering different periods. However, the difference observed in the magnitude of the trends during
2004-2014 compared to the results provided by Cusack et al. (2012) suggested that meteorology (in this case a
large increase in 2012; cf. Figure 2), changing from year to year, also determined the degree of comparability of
trends observed over different periods.
As reported in Fig. 2b, the $PM_{2.5}$ trends were linear for the three considered periods (1.2% < NL < 1.5%; cf. Table
1) and ME fits did not differ very much from MK fits (Fig. 2a). Following the MK and ME fits, the trends were
similar for the periods 2004 – 2014 and 2002 – 2014, whereas results were different for the period 2002 – 2010
(Cusack et al., 2012). MK test provided magnitudes of the trends of -1.87 %/yr (p<0.1) and -2.26 %/yr (p<0.05)
for 2004 – 2014 and 2002 – 2014, respectively. Conversely, the magnitude of the trend for the period 2002 –
2010 was higher in absolute values (-3.42 %/yr with p<0.01). Consequently, following the ME test, the
characteristic times ($\tau$) of the periods 2004 – 2014 (31.7 yr) and 2002 – 2014 (34.2 yr) were similar and both
were higher compared with $\tau_1$ calculated for the period 2002 – 2010 (19.3 yr). Consequently, the higher TR was
observed for the period 2002 – 2010 (34%) for which the highest slope was also observed following both MK
and ME tests. The RC was the lowest (9%) for the period 2002 – 2010 compared with 2004 – 2014 and 2002 –
2014 (15-16%). The differences observed in the magnitude of the trends and residual components for the three
periods are mostly due to the increase in the $PM_{2.5}$ concentrations observed in 2011-2012 when mean $PM_{2.5}$
reached around 15 $\mu g/m^3$ similar to the concentrations measured during 2002 – 2004 (around 14-16 $\mu g/m^3$).
Thus, over relatively short periods (9 -11 yr), the effects of just one meteorologically different year were clearly
visible. Despite this, we highlight that a statistically significant trend was observed thus confirming the
effectiveness of mitigation measures together with the effect of the economic crisis in Spain from 2008.

### 3.3 Trends of chemical species

The trends of the concentrations (using annual averages) of chemical species at BCN and MSY are reported in
Table 2 (for $PM_{10}$) and Table 3 (for $PM_{2.5}$). Figure 3 (for BCN) and Figure 4 (for MSY) show the trends of chemical
species in $PM_{10}$.
Here we assume that the change of the station in BCN in 2009 affected the trends of the concentrations of Cr,
Cu, Sn and Sb (traffic tracers), $Al_2O_3$, Ca, Mg, Ti, Rb, Sr (crustal elements related with both natural and
anthropogenic sources) and Fe (traffic and crustal tracer). These chemical species were highlighted in yellow in
Tables 2 and 3.
Figure 3 shows the impact of the change of the station for some of the aforementioned species in the year
2009. However, this change did not affect other species reported in Tables 2 and 3 and in Fig. 3, which had less
local character. These are $SO_4^{2-}$, $NH_4^+$, V, Ni (related with heavy oil combustion in the study area according to





source apportionment results, cf. Par. 4), Pb, Cd, As and Zn (related with industrial/metallurgy activities), Na and
Cl (sea spray), and $NO_3^-$. Although nitrate particles in Barcelona were mainly from traffic, the concentrations of
these particles were not strongly affected by the change of the station due to their secondary origin. For the
aforementioned reasons the comparison between BCN and MSY will be performed only using species (and PMF
sources) not affected by the change of the station in BCN. The MSY station will be considered as reference
station given that no location changes occurred at this monitoring site during the study period.
For the industrial tracers (Pb, Cd, As) the trends were similar in both PM fractions at both BCN and MSY.
Following the MK test, the decreasing trends in $PM_{10}$ fraction were statistically significant and ranged between -
3.43 %/yr (Cd and As; p<0.001) to -3.74 %/yr (Pb; p<0.001) at BCN and from -3.11 %/yr (Pb; p<0.01) to – 3.74
%/yr (As; p<0.001) at MSY. In $PM_{2.5}$ the magnitude of the trends were similar and ranged between -3.58 %/yr
(Cd; p<0.001) and -4.05 %/yr (Pb; p<0.001) at BCN and between -3.27 %/yr (As; p<0.01) and -3.74 %/yr (Pb;
p<0.001) at MSY. Similar magnitude of the trends for these species in both PM fractions at both sites confirmed
the common origin of these elements and the impact at regional scale of industrial sources.
However, it can be noted that the trends were not linear and the NL parameters were always higher than 10%
for Cd, Pb and As with the exception of As at MSY in $PM_{10}$ (cf. Tables 2 and 3). Thus, for these industrial tracers
single or double exponential fits were on average needed indicating that the trends were not gradual and
consistent over time and that the effectiveness of the control measures for these pollutants were stronger at
the beginning of the period under study (2004-2009 approximately) compared to the end of the period (Figs. 3
and 4). This is also evident by comparing the linear MK fit (dashed black line) with the ME fit (red lines) in Figs. 3
and 4. Note that at BCN a double exponential (DE) fit was needed to simulate the trends of Pb and Cd
concentrations in $PM_{10}$ likely because of the proximity of BCN measuring station to industrial emissions
compared to MSY station. For these industrial species in $PM_{10}$ one of the two characteristic times ($\tau$) was slightly
negative (cf. Tables 2) indicating both a slight increase of the simulated Pb and Cd concentrations in 2013-2014
and the strong decrease at the beginning of the period under study. Interestingly, as shown later, the $PM_{10}$
*Industrial/metallurgy* source contribution at BCN also showed a DE decreasing trend. The decrease observed for
Pb, Cd and As may be attributed to a decrease in the emissions from industrial production (smelters, Querol et
al., 2007) at a regional scale around Barcelona. The implementation of the IPPC Directive in 2008 in Spain is the
most probable cause for this downward trend. In $PM_{10}$ the TR from ME test ranged between 67% for As and
73% for Pb, at BCN, and from 48% for As and 72% for Cd, at MSY. In $PM_{2.5}$ TRs were similar to those observed in
$PM_{10}$ as expected given the general fine size mode of these industrial tracers (Table 1 and 2). The RC were quite
low and never exceeded 18% thus suggesting the goodness of the exponential fits used to study the trends of
the measured specie concentrations.
The comparison among the different ME trends in terms of characteristic times ($\tau$) is complicated by the fact
that $\tau$ is especially sensitive to the noise introduced by the inter-annual variability over the period considered
here (11 years). However, some interesting features can be observed. Characteristic times in $PM_{2.5}$ were very





similar for Pb and Cd at both stations (between 6.08 yr and 6.81 yr; cf. Table 3). For As in $PM_{2.5}$ the characteristic
times were similar between BCN (9.00 yr) and MSY (8.56 yr) but both were higher compared to Cd and Pb, due
to the slightly less intense exponential downward trend observed for As compared to Cd and Pb. Note that the
PMF analysis (cf. Paragraph 4) revealed that the concentrations of As were explained by multiple sources
(especially at BCN) whereas the *Industrial/metallurgy* source alone explained more than around 70% of Pb and
Cd concentrations (not shown). In $PM_{10}$ in Barcelona the characteristic decreasing times for Pb, Cd and As were
similar to those calculated at the same station in $PM_{2.5}$ (despite the different exponential fits used), whereas at
MSY the $PM_{10}$ characteristic times were slightly higher compared with $PM_{2.5}$ especially for As. It should be
considered that experimental uncertainties might also contribute to the observed differences in the
characteristic decreasing times over the period considered.  Moreover, different sampling days for $PM_{10}$ and
$PM_{2.5}$ together with possible sources of coarse As might also contribute to the observed differences.
In BCN the magnitude of the trend of $PM_{10}$ Zn was -3.74 % (p<0.001), from the MK test, and it was comparable
with the magnitudes estimated for As, Cd and Pb. However, following the ME test, the trend was linear (NL=6%)
and the total reduction was lower (TR=50%) compared to the other industrial tracers. In $PM_{2.5}$ the magnitude of
the trend of Zn was lower (-2.65 %/yr; p<0.01) compared with As, Cd and Pb and, again, the trend was linear
(NL=5%) and the total reduction (TR=46%) was low compared to the other industrial tracers. At MSY, Zn in $PM_{10}$
showed no statistically decreasing linear trend (-1.40 %/yr; NL = 1%) with low TR of around 23%. Thus $PM_{10}$ Zn
behaved clearly differently compared with As, Pb and Cd. The same was observed in $PM_{2.5}$ at MSY where the
magnitude of the trend for Zn (-2.02 %/yr; p<0.1 from MK test) was lower compared with As, Cd and Pb and
double exponential fit was needed following the ME test. As reported in Table 2, the two characteristics times
of the trend of Zn in $PM_{2.5}$ at MSY were positive and very much different. The exponent with lowest $\tau$ (0.21) was
needed to explain the strong decrease observed for $PM_{2.5}$ Zn between 2004 and 2005 (Figure SI-2). This
different behavior of Zn was likely due to the fact that Zn had multiple origins at both sites being related with
both industrial/metallurgy and road traffic emissions.
The concentrations of V and Ni in Barcelona in $PM_{10}$ showed similar trends decreasing at a rate of -3.11 %/yr
(p<0.001) and -3.43 %/yr (p<0.001), respectively, following the MK test. Both elements had single-exponential
decreasing trends (NL = 12% and 11% for V and Ni, respectively) with very similar decreasing times (10.04 yr and
10.61 yr, respectively), TR (63% and 61%, respectively) and RC (17% and 16%, respectively), thus conforming the
common origin of these two elements. In $PM_{2.5}$ in Barcelona, the magnitude of the decreasing trends, following
the MK test, were slightly lower compared with $PM_{10}$ (-2.96 %/yr and -3.11 %/yr for V and Ni, respectively;
p<0.01) and the trends were linear (NL = 9% for both elements). Characteristic times, TR and RC in $PM_{2.5}$ for V
and Ni in BCN were similar to those calculated in $PM_{10}$. Again we recall that the comparison of the fitting
parameters in $PM_{10}$ and $PM_{2.5}$ might be affected by different number of samples available for these PM size
fractions. At MSY, V and Ni showed similar trends as in BCN in both PM fractions. The concentrations of V
decreased at the rate of -3.27 %/yr (p<0.01; linear trend with NL = 7%) and -2.65 %/yr (SE trend with NL = 10%)
in $PM_{10}$ and $PM_{2.5}$, respectively. The concentrations of Ni decreased at the rate of -3.11 %/yr (p<0.01; linear



trend with NL = 5%) and -2.80 %/yr (SE trend with NL = 13%) in $PM_{10}$ and $PM_{2.5}$, respectively. Moreover, the
characteristic times $\tau$, TR and RC for V and Ni calculated at MSY were similar to those calculated at BCN.
As stated before the trends for Cr, Sn, Cu and Sb, typical traffic tracers, were studied only for MSY station as a
consequence of the change of the station in BCN in 2009. Sn and Cu in $PM_{10}$ at MSY showed very similar
behavior, both decreasing at the rate of -2.34 %/yr (p<0.05) with linear trends (2% < NL < 4%) and quite similar
TR ( 31-45%) and RC (15-19%). In $PM_{2.5}$, the concentrations of Sn and Cu at MSY showed marked trends
compared with $PM_{10}$, decreasing at -3.74 %/yr (p<0.001) and    -3.27 %/yr (p<0.01), respectively. In $PM_{2.5}$ the
trends were DE for Sn (see. Figure SI-3) and SE for Cu and TR (67-76%) and RC (9-13%) were similar for both
elements. The DE trend for $PM_{2.5}$ Sn at MSY was characterized (as for Zn) by a strong decrease between 2004
and 2005 and positive characteristic times (0.23 yr and 11.79 yr; cf. Figure SI-3). For Cr the situation was
different compared to Sn and Cu and in $PM_{10}$ at MSY Cr showed no trend (0.00 %/yr). In $PM_{2.5}$ a slight and not
statistically significant decreasing trend was detected for Cr concentrations(-1.18 %/yr; p>0.1) which was linear
following the ME fit.  In $PM_{2.5}$ the TR (31%) calculated for Cr was lower compared to Sn and Cu.
Sulfate ($SO_4^{2-}$) and ammonium ($NH_4^+$) particles concentrations showed very similar behavior in both $PM_{2.5}$ and
$PM_{10}$ size fractions, due to their fine nature, at both sites. In BCN the magnitude of the trends were -3.74 %/yr
(p<0.001) and -3.58 %/yr (p<0.001) for $SO_4^{2-}$ and $NH_4^+$, respectively. Both trends were SE with very similar
characteristic times (9.64-9.81 yr), NL (12%), TR (64-65%) and RC (12-14%). Very similar values were obtained
for these two species in $PM_{2.5}$ at BCN (cf. Table 3). At MSY both the magnitude of the trends of $SO_4^{2-}$ and $NH_4^+$
from MK test and their statistically significance were lower compared to BCN in both fractions. The trends were
linear for $SO_4^{2-}$ in both fractions and it was SE for $NH_4^+$. These differences could be explained by the distance of
MSY to direct specific sources of sulfate, such as shipping, compared to BCN, thus slightly reducing the
magnitude and the statistically significance of the trend of $SO_4^{2-}$ at regional level. Similar result was observed for
the contribution of the *Ammonium sulfate* source at both sites (cf. Table 5) with slightly higher magnitude of the
trend observed at BCN compared to MSY. Possible reasons for the observed reduction in the concentrations of
ambient sulfate in and around Barcelona will be discussed later.
Fine $NO_3^-$ (Table 3) showed very similar trends at both sites (-3.43 and -3.58 %/yr at MSY and BCN, respectively,
from MK test) with high statistical significance (p<0.001). At both stations $NO_3^-$ trends were SE and showed
similar $\tau$ (5.81 yr and 7.61 yr), TR (82% and 73%) and RC (21% and 16%). In $PM_{10}$ the trends of $NO_3^-$ at BCN and
MSY were similar to those observed in $PM_{2.5}$   (-3.11 %/yr with p<0.01 from MK test). Following the ME test the
TR for $PM_{10}$ $NO_3^-$ were lower (54% at MSY and 64% at BCN) compared to $PM_{2.5}$ and the characteristic times in
$PM_{10}$ were also lower compared to $PM_{2.5}$ at both sites. Thus, we deduced from ME analysis that fine $NO_3^-$ had a
more pronounced downward trend compared to $PM_{10}$ $NO_3^-$ and mainly at MSY where $\tau$ $PM_{2.5}$ and $\tau$ $PM_{10}$ were
5.81 yr and 12.71 yr, respectively. As reported in *Paragraph 4.1* statistically significant decreasing trends were
observed also for the *Ammonium Nitrate* source contributions at both sites. Possible reasons for the observed





decreases of $NO_3^-$ concentrations and *Ammonium Nitrate* source contributions will be discussed in *Paragraph.*
*4.1*.
As for Cr, Sn, Sb and Cu, the trends of mineral species ($Al_2O_3$, Ca, Fe) were studied only at MSY station. For these
elements, linear (with the exception of $Al_2O_3$ in $PM_{2.5}$ which was SE) and statistically significant decreasing
trends (with the exception of Ca in $PM_{2.5}$ with p>0.1) were detected. On average the TR were higher in the fine
fraction, ranging from 50% for Ca to 66% for $Al_2O_3$, compared to $PM_{10}$ (16-38% cf. Table 2) thus likely suggesting
a decrease with time of the concentrations of anthropogenic mineral species from specific sources such as
cement and concrete production and production works. In fact, coarse mineral matter at regional background
sites is mainly of natural origin. Downward decreasing trend for mineral matter contribution in $PM_{2.5}$ at MSY
was also reported by Cusack et al. (2012) for the period 2002 – 2010 at the same station. It is probable that
variations in meteorological conditions from one year to another (i.e. intensity and frequency of Saharan dust
outbreaks) might also explain the observed trend of mineral tracers at regional level.
The concentrations of $Cl^-$ did not show statistically significant trends in both fractions and at both sites due to its
main natural origin. Conversely, following the MK test, slightly statistical significant decreasing trends were
observed for Na at both sites and in both fractions with the exception of Na in $PM_{10}$ at MSY for which no trend
was detected.
Finally, at MSY neither OC nor EC concentrations showed statistically significant trends (not shown). Consider
that the concentrations of EC at MSY are very low and around at 0.2-0.3 $\mu g/m^3$ as annual mean. Both
anthropogenic activity and biomass burning were expected to contribute to this chemical specie. Concerning OC
the lack of trend was probably due to the contribution from biogenic sources (and meteorology) to the
concentration of this specie at regional level.

**4. PMF source profiles and contributions**
Eight and seven sources were detected at BCN and MSY, respectively, in $PM_{10}$ from PMF model. The absolute
and relative contributions of these sources to the measured $PM_{10}$ mass are reported in Figure 5 and Table 4.
The chemical profiles of the detected sources were reported in Supporting Information (Figure SI-4).
Some of these sources were common at both BCN and MSY. These are: *Ammonium Sulfate* (secondary inorganic
source traced by $SO_4^{2-}$ and $NH_4^+$ and contributing 3.95 $\mu g/m^3$ (23.7%) and 4.67 $\mu g/m^3$ (13.7%) at MSY and BCN,
respectively), *Ammonium nitrate* (secondary inorganic source traced by $NO_3^-$ and $NH_4^+$ and contributing 1.31
$\mu g/m^3$ (7.9%) and 4.45 $\mu g/m^3$ (13.1%) at MSY and BCN, respectively), *V-Ni bearing* source (traced mainly by V, Ni
and $SO_4^{2-}$ it represents the direct emissions from heavy oil combustion and contributed 0.71 $\mu g/m^3$ (4.3%) and
3.32 $\mu g/m^3$ (9.8%) at MSY and BCN, respectively), *Mineral* (traced by typical crustal elements such as Al, Ca, Ti,
Rb, Sr and contributing 2.70 $\mu g/m^3$ (16.2%) and 4.61 $\mu g/m^3$ (13.6%) at MSY and BCN, respectively), *Aged marine*



(traced by Na and Cl mainly with contributions from $SO_4^{2-}$ and $NO_3^-$ and contributing 1.76 μg/m$^3$ (10.6%) and
5.73 μg/m$^3$ (16.9%) at MSY and BCN, respectively). Sources detected at MSY but not at BCN were:
*Industrial/Traffic* source (traced by EC, OC, Cr, Cu, Zn, As, Cd, Sn, Sb and Pb it includes mixed contributions from
anthropogenic sources such as road traffic and metallurgic industries and contributed 1.43 μg/m$^3$ (8.6%)) and
*Aged organics* (traced mainly by OC and EC with maxima in summer indicating mainly a biogenic origin and
contributing 3.78 μg/m$^3$ (22.7%)). The ratio OC:EC in the *Industrial/Traffic* and *Aged organic* source profiles at
MSY were 4.2 and 11.7, respectively, thus indicating a strong influence of aged particles in the latter source with
the former source being more fresh. The statistic of the OC:EC ratio based on chemical data at MSY is reported
in Supporting Information (Figure SI-5). Mean and median values of OC:EC ratio at MSY were 9.1 and 7.8,
respectively.
Finally, some sources were detected at BCN but not at MSY: *traffic* (traced by $C_{nm}$, Cr, Cu, Sb and Fe mainly and
contributing 5.14 μg/m$^3$ (15.1%)), *road/work resuspension* (traced by both crustal elements, mainly Ca, and
traffic tracers such as Sb, Cu and Sn and contributing 4.25 μg/m$^3$ (12.5%)) and *Industrial/metallurgy* (traced by
Pb, Cd, As and Zn and contributing 0.96 μg/m$^3$ (2.8%).
A sensitivity study was performed in order to better interpret the PMF sources at BCN. In fact, for the period
2007 – 2014 separate OC and EC concentration measurements were available and a PMF was performed. The
comparison between the PMF source contributions obtained using the period 2007-2014 (separate OC and EC
measurements) and the whole period (2004-2014; $C_{nm}$ (non-mineral carbon) available) is reported in Supporting
Information (Figure SI-6). As reported in Figure SI-6 the differences in source contribution and R$^2$ ranged
between -3% (*Mineral* source) and +20% (*Industrial* source) and 0.894 to 0.997, respectively, thus confirming
the correct interpretation of the 2004-2014 PMF sources where $C_{nm}$ was used. The OC:EC ratio in the *Traffic*
source from 2007-2014 PMF was 1.70 (cf. Figure SI-7) whereas the mean and median OC:EC ratio from
chemistry data were 2.5 and 2.3, respectively, thus being in agreement with the contribution of fresh particles
from *Traffic* source at BCN.

### 4.1 Trends of PM$_{10}$ source contributions: annual averages

Figures 6 and 7 and Table 5 show the results from MK and ME test applied to the annual averages of PM$_{10}$
source contributions at BCN and MSY. Following the MK test the source contributions that showed statistically
significant downward trends at both stations were *Ammonium sulfate*, *Ammonium nitrate*, and *V-Ni bearing*
sources. Moreover, statistically significant decreasing trends were observed for the *Industrial/Traffic* source
(including traffic and industrial emissions at MSY) and the *Industrial/metallurgy* source (at BCN). These sources
were clearly linked with anthropogenic activities and the observed decreasing trends confirmed the
effectiveness of pollution control measures discussed before and the possible effect of the economic crisis. The
contributions of these four sources at BCN were not affected by the change of the station in 2009 and,



consequently, are discussed here. However, as already noted we cannot study trends for *Traffic*, *Road/work*
*resuspension* and *Mineral* source contributions at BCN.
Consistently with the trends of $SO_4^{2-}$, $NO_3^-$ and $NH_4^+$ (Table 2), *Ammonium sulfate* and *Ammonium nitrate* source
contributions in $PM_{10}$ showed highly statistically significant decreasing trends at both stations (Table 5). The
trends from MK test were -2.49 %/yr (p<0.05) and -3.74 %/yr (p<0.001) for *Ammonium sulfate* and *Ammonium*
*nitrates* contributions, respectively, at BCN and -2.02 %/yr (p<0.05) and -3.27 %/yr (p<0.01), respectively, at
MSY. At both stations the trend of the contribution of the *Ammonium sulfate* source to $PM_{10}$ was DE (Double
Exponential) with two positive characteristic times ($\tau$; Table 5) one of which was lower than the other
(especially at BCN) indicating a stronger decrease at the beginning of the considered period (cf. Figures 6 and 7).
Thus, the decrease over time of the contribution from the *Ammonium sulfate* source was not gradual and
monotonic. Note that at MSY there was a small statistically significant difference between SE and DE fits.
However, the DE fit was chosen reducing considerably the RC for this source. The TR was quite similar at both
sites: 67% at BCN and 51% at MSY and the RC was low at both sites (16-21%) again demonstrating the goodness
of the used fits. The observed decrease in the *Ammonium sulfate* source contribution may be attributed to the
legislation that came into force in 2007-2008 in Spain, the EC Directive on Large Combustion Plants, which
resulted in the application of flue gas desulfurization (FGD) systems in a number of large facilities in 2007-2008
in Spain. Moreover, from 2008 on, the use of heavy oils and petroleum coke for power generation was
forbidden around Barcelona, thus only natural gas was allowed to this end according to the 2008 Regional AQ
Plan.
*Ammonium nitrate* contribution trends were SE (Single Exponential) at both stations with very similar $\tau$ (8.96 yr
– 8.59 yr), TR (67-69 %) and RC (13-17%). Moreover, it should be noted that the lower MK magnitude (and
statistical significance) of the *Ammonium sulfate* contribution trends compared to the trends of the *Ammonium*
*nitrate* contributions was confirmed by the different fitting lines needed to fit these trends: double exponential
for *Ammonium sulfate* contributions and single exponential for *Ammonium* nitrate contributions. Thus, the
contributions to $PM_{10}$ from the *Ammonium nitrate* source showed a more gradual and consistent over time
decrease compared to the contributions from the *Ammonium sulfate* source.
The decreases observed for the concentrations of nitrates ($NO_3^-$; cf. Par. 3.2) and for the contribution from the
*Ammonium Nitrate* source were mainly related to the reduction in ambient $NO_x$ concentrations. Figure 8 shows
the levels of $NO_2$ from 2005 to 2014 in South Europe from NASA $NO_2$ OMI level3 plotted using the Giovanni
online data system (Acker and Leptoukh, 2007). In Spain it can be observed a general decrease of the
concentrations of $NO_2$ at regional level due to lower energy consumption mainly related with the financial crisis.
The decrease of $NO_3^-$ concentrations and *Ammonium nitrate* source contributions around Barcelona was
attributed to the decrease of $NO_x$ emissions mainly from the five power generation plants around the city.
Moreover, during 2008-2012 the Regional AQ Plan has driven the implementation of SCRT (continuously





regenerating PM traps with selective catalytic reduction for $NO_2$) and the hybridization and shift to natural gas
engines of the Barcelona´s bus fleet may have had also an influence in $NO_x$ ambient concentrations.
The decreasing trends of the *V-Ni bearing* source from MK analysis were similar at both stations and around -
3.11 %/yr at BCN (p<0.01) and -2.96 %/yr (p<0.01) at MSY. The trends were linear (L) at MSY (NL=8%) and SE at
BCN (NL=11%). Very similar characteristic times ($\tau$) were observed at both stations for the *V-Ni bearing* source
contributions: 10.59 yr at BCN and 11.94 yr at MSY. Interestingly, the $\tau$ calculated for the V-Ni bearing source
were very similar to those calculated for V and Ni species in $PM_{10}$ (cf. Table 2).  The *V-Ni bearing* TR was around
61% at BCN and 57% at MSY and RCs were similar (19-25 %). Note that following the MK test the magnitude of
the trends for these three sources were on average higher at BCN compared to MSY likely due to the proximity
of the BCN measuring station to anthropogenic sources of V and Ni (mainly shipping). The observed decrease in
the *V-Ni bearing* source contribution was mainly attributed to the ban of the use of heavy oils and petroleum
coke for power generation from 2008.
The *Industrial/Metallurgy* source contribution at BCN decreased at the rate of -3.43 %/yr from MK test
(p<0.001).  As already observed this decrease was mainly attributed to the implementation of IPPC Directives.
Moreover, the observed decrease may be attributed to a decrease in the emissions from industrial production
(smelters, Querol et al., 2007) at a regional scale around Barcelona. The implementation of the IPPC Directive in
2008 in Spain was the most probable cause for this downward trend. Also for this source the trend was DE with
one negative characteristic time (cf. Table 5) due to a slight increase of the *Industrial/Metallurgy* source
contribution at the end of the considered period. This was consistent with the trends observed for the *Industrial*
tracers at BCN (cf. Figure 3 and Table 2). TR and RC for the *Industrial* source contributions at BCN were 56% and
11%, respectively. Note again that low RC indicates a good fit of the data using the multi-exponential approach.
Interestingly, the contribution of the *Industrial/Traffic* source at MSY had a very similar magnitude of the trend
(-3.11 %/yr from MK test with p<0.01) compared to the BCN *Industrial* contribution trend, being both sources
traced mostly by the same industrial tracers. Also the TR and RC were similar (54% and 11%, respectively at
MSY).
At MSY the contribution of the *Aged organics* source increased at the rate of +1.25 %/yr however the trend was
not statistically significant. The increasing trend for this source was mainly due to the observed increase in the
contribution in the year 2011 when mean annual OC concentration at MSY reached a maximum in the
considered 11-yr period (not shown). The *Aged organic* source contribution (likely mainly driven by biogenic
POA and SOA formed in these forested regional area in the WMB, cf. Pandolfi et al., 2014) did not show
statistically significant trends being these contributions mainly driven by meteorology.
The *Mineral* source contribution at MSY showed statistically significant decreasing trend (p<0.1) whereas the
M*arine* source contributions showed no-statistically significant trends at both stations this being consistent with
the natural origin of this latter source. The statistically significant decreasing trend observed at MSY for the
*Mineral* source was in agreement with what observed at the same station by Cusack et al. (2012). This





decreasing trend could be due to a possible decrease of the emissions of anthropogenic mineral species from
specific sources such as cement and concrete production and construction works.
Finally, the possible relationship of the ambient air pollution trends with the emission patterns by sectors is
evaluated here using the NECo statistical data for Spain (MINETUR, 2013) as well as of the time trends for $SO_2$
and $NO_x$ emissions evidenced for the NEIs (MAGRAMA, 2013). The primary energy consumption increased by
+4% from 2004 to 2007, slightly decreased from 2007 to 2008 (−4%), and markedly decreased in 2009 (−12%
with respect to 2007) (Fig. 9). Since 2009, the energy consumption indicator seems to remain constantly low
until 2012 with an additional decrease in 2013 and 2014 (around -8% compared to 2009). Oil consumption was
fairly constant during 2004–2007 showing an important (−29%) decrease during 2007–2014, without major
changes from 2007 to 2008 (−3%). This trend is probably governed by the fuel consumption for traffic road.
Natural gas consumption increased (+39%) from 2004 to 2008, and then diminished by -45% in 2014 compared
to 2008. Coal consumption remained constantly high from 2004 to 2007 whereas, as for the emissions of $SO_2$
(Fig. 10), a sharp decrease occurred from 2007 to 2008 (−31%), continuing until 2010 (−61% compared to 2007).
However, in the period 2011-2014 there was an important increase leading to an average consumption similar
to the consumption for the year 2008. Nuclear energy consumption remained relatively constant along the
study period, whereas the hydroelectric generation had three maxima in 2010, 2013 and 2014, coinciding with
the fall of coal consumption. The 2010 increase of hydroelectric consumption was very remarkable and due to
the high rainfall rate of this year. Thus, 2010 was a favorable year for atmospheric dispersion and washout of
pollution in Spain, but air quality also improved by lower atmospheric emissions due to the decrease of coal
consumption in favor of a hydroelectric growth. Renewable energy consumption increased by 440% from 2004
to 2014, with a gradual growth in the NECo.

**5.0 Conclusions**

PM chemical speciated data collected at two twin stations in NE of Spain (one urban background station and
one regional background station) during 2004 – 2014 were used to study trends of source contributions from
PMF analysis and of chemical species concentrations. Despite the fact the trends of different PM fractions ($PM_1$,
$PM_{2.5}$ and $PM_{10}$) were linear during the period under study, the trends of specific chemical elements and source
contributions were single or double exponential demonstrating the different effectiveness and time of
implementation of different reduction strategies on specific pollutant sources. The contributions that showed
statistically significant downward trends at both Barcelona (BCN; UB) and Montseny (MSY; RB) were from
*Ammonium sulfate*, *Ammonium nitrate*, and *V-Ni bearing* sources. Moreover, statistically significant decreasing
trends were observed for the *Industrial/Traffic* source (at MSY; mixed road traffic and metallurgy) and the
*Industrial/metallurgy* source (at BCN). These sources were clearly linked with anthropogenic activities and the
observed decreasing trends confirmed the effectiveness of pollution control measures implemented at EU or
regional/local levels. Moreover, the economic crisis which started in 2008 in Spain also contributed to the




observed trends. The general trends observed for the calculated PMF source contributions well reflected the
trends observed for the chemical tracers of these pollutant sources. At both sites the decreasing trends of the
*Ammonium sulfate* source contributions were double exponential indicating that the trends were not gradual
and consistent with time. In fact, the observed decrease in the *Ammonium sulfate* source contribution was
mainly attributed to the EC Directive on Large Combustion Plants implemented from 2008 in Spain, resulting in
the application of fuels gas desulfurization (FGD) systems in a number of large facilities. Moreover, the use of
heavy oils and petroleum coke for power generation was forbidden around Barcelona from 2008, thus only
natural gas was allowed to this end according to the 2008 Regional AQ Plan. Conversely, *Ammonium nitrate* and
*V-Ni bearing* source contributions were well fitted by a single exponential curve suggesting a more gradual and
consistent with time decreasing trends for these contributions compared to *Ammonium sulfate* source
contribution. The decrease observed for the contribution of the *Ammonium Nitrate* source was mainly due to
the reduction in ambient $NO_x$ concentrations. In Spain a general decrease of the concentrations of $NO_2$ at
regional level was observed and it was mainly related with the lower energy consumption related with the
financial crisis. The decrease of nitrates concentrations and *Ammonium nitrate* source contributions around
Barcelona was also attributed to the decrease of $NO_x$ emissions from the five power generation plants around
the city. Moreover, a Regional AQ Plan implementing the SCRT (continuously regenerating PM traps with
selective catalytic reduction for $NO_2$) and the hybridization and shift to natural gas engines of the Barcelona´s
bus fleet may have had also an influence in $NO_x$ ambient concentrations. The magnitude of the decreasing
trends of the contributions of the aforementioned sources were always higher at BCN compared to MSY likely
because of the proximity of the BCN measurement site to anthropogenic pollutant sources compared to the
MSY site.

**Acknowledgments.**
This work was supported by the MINECO (Spanish Ministry of Economy and Competitiveness), the MAGRAMA
(Spanish Ministry of Agriculture, Food and Environment), the Generalitat de Catalunya (AGAUR 2014 SGR33 and
the DGQA) and FEDER funds under the PRISMA project (CGL2012-39623- C02/00). The research leading to these
results has received funding from the European Union's Horizon 2020 research and innovation programme
under grant agreement No 654109 and previously from the European Union Seventh Framework Programme
(FP7/2007-2013) under grant agreement n° 262254. $NO_2$ map analyses and visualizations used in this paper
were produced with the Giovanni online data system, developed and maintained by the NASA GES DISC. The
authors would like to express their gratitude to D. C. Carslaw and K. Ropkins for providing the Openair software
used in this paper (Carslaw and Ropkins, 2012; Carslaw, 2012).




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




**Table 1**: Mann-Kendall and Multi-exponential trends of different PM fractions from both gravimetry (grav) and optical (OPC)
measurements at BCN (bold italic) and MSY (2004-2014). Type: linear (L), single-exponential (SE), double exponential (DE); a ($\mu$gm$^{-3}$) and
$\tau$ (yr) are the constant and the characteristic time, respectively, of the exponential fit of the data; NL (%) = Non-Linearity; TR (%) = Total
Reduction; RC (%) = Residual Component. Significance of the trends following the Mann-Kendall test: *** (p-value < 0.001), ** (p-value <
0.01), * (p-value < 0.05), + (p-value < 0.1).

| PM$_x$ | PM$_x$ | | Mann-Kendall fit | | Multi-exponential fit | | | | | |
|---|---|---|---|---|---|---|---|---|---|---|
| Fraction | Conc. 2004 ($\mu$gm$^{-3}$) | Conc. 2014 ($\mu$gm$^{-3}$) | Trend (%/yr) | p-value | type | a ($\mu$gm$^{-3}$) | $\tau$ (yr) | NL (%) | TR (%) | RC (%) |
| PM$_{10}$ (grav) | *41.12* | *19.16* | *-3.58* | *** | *L* | *50.30* | *11.60* | *9* | *58* | *10* |
| | 19.24 | 13.88 | -0.47 | | L | 17.92 | 79.46 | 0 | 12 | 17 |
| PM$_{2.5}$ (grav) | *31.61* | *13.18* | *-3.89* | *** | *L* | *32.41* | *10.87* | *9* | *60* | *8* |
| | 16.18 | 9.80 | -1.87 | + | L | 14.36 | 31.70 | 1 | 27 | 16 |
| PM$_{10}$ (OPC) | *39.06* | *19.84* | *-3.43* | *** | *L* | *45.85* | *15.62* | *5* | *47* | *11* |
| | 18.63 | 12.34 | 0.00 | | L | 16.25 | 115.92 | 0 | 8 | 15 |
| PM$_{2.5}$ (OPC) | *27.12* | *12.86* | *-3.11* | ** | *L* | *29.33* | *15.51* | *5* | *53* | *11* |
| | 15.71 | 9.28 | -1.78 | + | L | 12.46 | 40.59 | 0 | 24 | 14 |
| PM$_1$ (OPC) | *20.78* | *9.05* | *-3.11* | ** | *L* | *23.03* | *13.83* | *6* | *51* | *13* |
| | 12.86 | 6.68 | -1.25 | | L | 10.90 | 37.33 | 1 | 23 | 17 |
| PM$_{1-10}$ (OPC) | *18.29* | *11.48* | *-2.34* | * | *L* | *22.86* | *17.92* | *4* | *43* | *16* |
| | 5.94 | 5.51 | 0.93 | | L | 5.61 | -68.77 | 0 | -16 | 23 |
| PM$_{1/10}$ (OPC) | *0.55* | *0.46* | *-0.31* | | *L* | *0.50* | *135.80* | *0* | *7* | *9* |
| | 0.71 | 0.59 | -1.25 | | L | 0.67 | 70.88 | 0 | 13 | 10 |
| PM$_{2.5-10}$ (OPC) | *13.07* | *7.89* | *-2.18* | * | *L* | *14.72* | *16.25* | *5* | *46* | *18* |
| | 4.09 | 3.79 | 0.62 | | L | 3.88 | -76.32 | 0 | -14 | 26 |
| PM$_{2.5/10}$ (OPC) | *0.66* | *0.63* | *-0.62* | | *L* | *0.68* | *396.60* | *0* | *2* | *4* |
| | 0.80 | 0.77 | -0.62 | | L | 0.76 | 288.49 | 0 | 3 | 9 |














**Table 2**: Mann-Kendall and Multi-exponential trends of different chemical species in $PM_{10}$ at BCN (bold italic) and MSY. Colour highlights
species in BCN for which trends cannot be studied due to the change of the BCN station in 2009. Type: linear (L), single-exponential (SE),
double exponential (DE); **a** ($\mu gm^{-3}$) and $\tau$ (yr) are the constants and the characteristic times, respectively, of the exponential data fittings;
NL (%) = Non-Linearity; TR (%) = Total Reduction; RC (%) = Residual Component. Significance of the trends following the Mann-Kendall
test: *** (p-value < 0.001), ** (p-value < 0.01), * (p-value < 0.05), + (p-value < 0.1).

| Specie | $PM_{10}$ (*BCN*;MSY) Concentration 2004 ($\mu gm^{-3}$) | Concentration 2014 ($\mu gm^{-3}$) | Mann-Kendall fit Trend (%/yr) | p-value | Multi-exponential fit type | a ($\mu gm^{-3}$) | $\tau$ (yr) | NL (%) | TR (%) | RC (%) |
|---|---|---|---|---|---|---|---|---|---|---|
| Pb | *0.02685* | *0.00694* | *-3.74* | *** | *DE* | *0.03409* *1.64E-5* | *5.38* *-2.09* | *39* | *73* | *10* |
| | 0.00481 | 0.00190 | -3.11 | ** | SE | 0.00553 | 10.22 | 11 | 62 | 13 |
| Cd | *0.00043* | *0.00015* | *-3.43* | *** | *DE* | *0.00051* *3.36E-8* | *6.64* *-1.48* | *33* | *65* | *11* |
| | 0.00017 | 0.00006 | -3.27 | ** | SE | 0.00018 | 7.92 | 18 | 72 | 16 |
| As | *0.00094* | *0.00036* | *-3.43* | *** | *SE* | *0.00118* | *9.11* | *14* | *67* | *11* |
| | 0.00029 | 0.00017 | -3.74 | *** | L | 0.00031 | 15.39 | 5 | 48 | 10 |
| Zn | *0.10584* | *0.05483* | *-3.74* | *** | *L* | *0.11310* | *14.49* | *6* | *50* | *7* |
| | 0.01401 | 0.00912 | -1.40 | | L | 0.01434 | 38.46 | 1 | 23 | 19 |
| V | *0.01116* | *0.00454* | *-3.11* | ** | *SE* | *0.01502* | *10.04* | *12* | *63* | *17* |
| | 0.00328 | 0.00175 | -3.27 | ** | L | 0.00412 | 13.00 | 7 | 54 | 16 |
| Ni | *0.00531* | *0.00284* | *-3.43* | *** | *SE* | *0.00678* | *10.61* | *11* | *61* | *16* |
| | 0.00155 | 0.00100 | -3.11 | ** | L | 0.00180 | 15.73 | 5 | 47 | 21 |
| Cr | *0.00499* | *0.00316* | *-2.49* | * | *L* | *0.00641* | *15.38* | *5* | *48* | *18* |
| | 0.00102 | 0.00110 | 0.00 | | L | 0.00100 | 127.74 | 0 | 8 | 23 |
| Sn | *0.00532* | *0.00402* | *-2.02* | * | *L* | *0.00745* | *24.55* | *2* | *33* | *15* |
| | 0.00127 | 0.00057 | -2.34 | * | L | 0.00107 | 16.97 | 4 | 45 | 15 |
| Cu | *0.07064* | *0.01394* | *-2.96* | ** | *SE* | *0.10238* | *6.18* | *27* | *80* | *30* |
| | 0.00420 | 0.00216 | -2.34 | * | L | 0.00416 | 26.63 | 2 | 31 | 19 |
| Sb | *0.00894* | *0.00207* | *-2.96* | ** | *SE* | *0.01140* | *6.19* | *27* | *80* | *25* |
| | 0.00058 | 0.00025 | -2.80 | ** | SE | 0.00064 | 10.46 | 11 | 62 | 13 |
| $SO_4^{2-}$ | *5.74436* | *2.28596* | *-3.74* | *** | *SE* | *6.56033* | *9.81* | *12* | *64* | *12* |
| | 2.84849 | 1.67712 | -2.80 | ** | L | 3.00501 | 18.54 | 4 | 42 | 19 |
| $NO_3^-$ | *5.07816* | *1.72401* | *-3.11* | ** | *SE* | *6.49890* | *9.83* | *12* | *64* | *15* |
| | 1.80724 | 0.67419 | -3.11 | ** | L | 2.10077 | 12.71 | 7 | 54 | 14 |
| $NH_4^+$ | *1.92062* | *0.57008* | *-3.58* | *** | *SE* | *1.90645* | *9.64* | *12* | *65* | *14* |
| | 1.14268 | 0.40135 | -2.49 | * | SE | 1.28868 | 9.26 | 13 | 66 | 22 |
| $Al_2O_3$ | *1.25631* | *0.42353* | *-2.34* | * | *SE* | *1.83628* | *8.80* | *15* | *68* | *33* |
| | 0.72357 | 0.46382 | -2.18 | * | L | 0.71956 | 23.71 | 2 | 35 | 19 |
| Ca | *2.34383* | *0.49357* | *-2.96* | ** | *SE* | *3.35120* | *7.27* | *21* | *75* | *34* |
| | 0.42703 | 0.28279 | -2.49 | * | L | 0.47048 | 21.07 | 3 | 38 | 17 |
| Fe | *0.94101* | *0.46700* | *-2.18* | * | *L* | *1.21322* | *11.22* | *9* | *59* | *17* |
| | 0.22371 | 0.14895 | -1.71 | + | L | 0.22198 | 56.84 | 0 | 16 | 40 |
| Na | *1.02188* | *0.77408* | *-2.34* | * | *L* | *1.13649* | *28.31* | *2* | *30* | *13* |
| | 0.31184 | 0.33089 | 0.00 | | L | 0.33531 | 493.40 | 0 | 2 | 21 |
| Cl | *0.85827* | *0.78893* | *-1.09* | | *L* | *0.92544* | *40.26* | *1* | *22* | *20* |
| | 0.17991 | 0.34437 | 0.62 | | L | 0.22167 | -36.01 | 1 | -32 | 61 |






**Table 3**: Mann-Kendall and Multi-exponential trends of different chemical species in PM$_{2.5}$ at BCN (bold italic) and MSY. Colour highlights
species in BCN for which trends cannot be studied due to the change in the BCN station in 2009. Type: linear (L), single-exponential (SE),
double exponential (DE); **a** ($\mu$gm$^{-3}$) and $\tau$ (yr) are the constants and the characteristic times, respectively, of the exponential data fittings;
NL (%) = Non-Linearity; TR (%) = Total Reduction; RC (%) = Residual Component. Significance of the trends following the Mann-Kendall
test: *** (p-value < 0.001), ** (p-value < 0.01), * (p-value < 0.05), + (p-value < 0.1).

| Specie | PM$_{2.5}$ (*BCN*;MSY) Concentration 2004 ($\mu$gm$^{-3}$) | Concentration 2014 ($\mu$gm$^{-3}$) | Mann-Kendall fit Trend (%/yr) | p-value | Multi-exponential fit type | a ($\mu$gm$^{-3}$) | $\tau$ (yr) | NL (%) | TR (%) | RC (%) |
|---|---|---|---|---|---|---|---|---|---|---|
| Pb | *0.02117* | *0.00500* | *-4.05* | *** | *SE* | *0.02390* | *6.24* | *27* | *80* | *13* |
|  | 0.00642 | 0.00149 | -3.74 | *** | SE | 0.00716 | 6.08 | 28 | 81 | 18 |
| Cd | *0.00041* | *0.00011* | *-3.58* | *** | *SE* | *0.00047* | *6.81* | *23* | *77* | *13* |
|  | 0.00020 | 0.00005 | -3.58 | *** | SE | 0.00020 | 6.77 | 23 | 77 | 18 |
| As | *0.00069* | *0.00027* | *-3.74* | *** | *SE* | *0.00091* | *9.00* | *14* | *67* | *11* |
|  | 0.00029 | 0.00013 | -3.27 | ** | SE | 0.00033 | 8.56 | 15 | 69 | 19 |
| Zn | *0.07500* | *0.03985* | *-2.65* | ** | *L* | *0.07330* | *16.41* | *5* | *46* | *13* |
|  | 0.02649 | 0.01017 | -2.02 | * | DE | 1.53281 0.01377 | 0.21 22.56 | 49 | 68 | 18 |
| V | *0.00823* | *0.00368* | *-2.96* | ** | *L* | *0.01121* | *11.13* | *9* | *59* | *16* |
|  | 0.00271 | 0.00130 | -2.65 | ** | SE | 0.00338 | 10.77 | 10 | 60 | 21 |
| Ni | *0.00402* | *0.00185* | *-3.11* | ** | *L* | *0.00498* | *11.23* | *9* | *59* | *15* |
|  | 0.00189 | 0.00080 | -2.80 | ** | SE | 0.00205 | 9.36 | 13 | 42 | 24 |
| Cr | *0.00226* | *0.00155* | *-1.40* |  | *L* | *0.00297* | *16.05* | *5* | *46* | *27* |
|  | 0.00077 | 0.00062 | -1.18 |  | L | 0.00104 | 16.75 | 4 | 31 | 24 |
| Sn | *0.00268* | *0.00188* | *-2.34* | * | *L* | *0.00398* | *19.87* | *3* | *40* | *16* |
|  | 0.00157 | 0.00043 | -3.74 | *** | DE | 0.04830 0.00100 | 0.23 11.79 | 44 | 76 | 9 |
| Cu | *0.02747* | *0.00503* | *-3.11* | ** | *SE* | *0.05273* | *5.32* | *34* | *85* | *43* |
|  | 0.00394 | 0.00113 | -3.27 | ** | SE | 0.00426 | 8.99 | 14 | 67 | 13 |
| Sb | *0.00300* | *0.00078* | *-2.96* | ** | *DE* | *0.00395* *1.6E-6* | *5.10* *-1.98* | *42* | *73* | *18* |
|  | 0.00053 | 0.00015 | -2.80 | ** | DE | 0.00069 1.3E-6 | 4.52 -2.50 | 48 | 70 | 16 |
| SO$_4^{2-}$ | *4.86564* | *1.92388* | *-3.74* | *** | *SE* | *5.64582* | *9.69* | *12* | *64* | *9* |
|  | 2.98922 | 1.43381 | -2.96 | ** | L | 3.29195 | 11.70 | 9 | 57 | 16 |
| NO$_3^-$ | *3.45513* | *0.86002* | *-3.58* | *** | *SE* | *4.14459* | *7.61* | *19* | *73* | *16* |
|  | 1.66095 | 0.29452 | -3.43 | *** | SE | 1.96014 | 5.81 | 30 | 82 | 21 |
| NH$_4^+$ | *2.19735* | *0.68393* | *-3.58* | *** | *SE* | *2.27813* | *10.53* | *11* | *61* | *15* |
|  | 1.39366 | 0.48049 | -3.11 | ** | SE | 1.62588 | 7.94 | 18 | 72 | 14 |
| Al$_2$O$_3$ | *0.46954* | *0.14567* | *-2.65* | ** | *SE* | *0.73949* | *7.46* | *20* | *74* | *39* |
|  | 0.30245 | 0.10153 | -2.34 | * | SE | 0.26678 | 9.36 | 13 | 66 | 35 |
| Ca | *0.65421* | *0.13737* | *-2.80* | ** | *SE* | *1.09283* | *5.91* | *29* | *82* | *46* |
|  | 0.11478 | 0.06540 | -1.56 |  | L | 0.10688 | 14.36 | 6 | 50 | 33 |
| Fe | *0.32489* | *0.15007* | *-1.87* | + | *SE* | *0.45302* | *8.26* | *16* | *70* | *25* |
|  | 0.09679 | 0.03716 | -2.02 | * | L | 0.08842 | 12.20 | 8 | 56 | 31 |
| Na | *0.27476* | *0.17863* | *-1.87* | + | *L* | *0.32279* | *19.56* | *3* | *40* | *15* |
|  | 0.13091 | 0.07252 | -2.49 | * | L | 0.13516 | 18.56 | 4 | 42 | 19 |
| Cl | *0.37296* | *0.32756* | *-0.47* |  | *L* | *0.37806* | *23.25* | *2* | *35* | *49* |
|  | 0.10917 | 0.18225 | 0.00 |  | L | 0.16170 | -1029.00 | 0 | -1 | 79 |








**Table 4**: Source contributions at Barcelona (BCN) and Montseny (MSY)

|  | BCN | MSY |
|---|---|---|
|  | [µg/m³; %] | [µg/m³; %] |
| *Source* |  |  |
| AgedMarine | 5.73; **16.9** | 1.76; **10.6** |
| Mineral | 4.61; **13.6** | 2.70; **16.2** |
| AmmSulfate | 4.67; **13.7** | 3.95; **23.7** |
| AmmNitrate | 4.45; **13.1** | 1.31; **7.9** |
| V-Ni | 3.32; **9.8** | 0.71; **4.3** |
| Industrial/Metallurgy | 0.96; **2.8** |  |
| Traffic | 5.14; **15.1** |  |
| Road/work resuspension | 4.25; **12.5** |  |
| Aged Organics |  | 3.78; **22.7** |
| Industrial/Traffic |  | 1.43; **8.6** |
























**Table 5**: Mann-Kendall and Multi-exponential trends of source contributions in PM$_{10}$ from PMF at BCN (bold italic) and MSY. Highlighted
source contributions at BCN from Mineral, Traffic and Road/work resuspension were excluded from the trend discussion. Type: linear (L),
single-exponential (SE), double exponential (DE); **a** ($\mu$gm$^{-3}$) and $\tau$ (yr) are the constants and the characteristic times, respectively, of the
exponential data fittings; NL (%) = Non-Linearity; TR (%) = Total Reduction; RC (%) = Residual Component. Significance of the trends
following the Mann-Kendall test: *** (p-value < 0.001), ** (p-value < 0.01), * (p-value < 0.05), + (p-value < 0.1).

| Source | PM$_{10}$ (**BCN**;MSY) | | Mann-Kendall fit | | Multi-exponential fit | | | | | |
|---|---|---|---|---|---|---|---|---|---|---|
| | Contribution 2004 ($\mu$gm$^{-3}$) | Contribution 2014 ($\mu$gm$^{-3}$) | Trend (%/yr) | p-value | type | a ($\mu$gm$^{-3}$) | $\tau$ (yr) | NL (%) | TR (%) | RC (%) |
| *Ammonium sulfate* | **10.27** | **3.38** | **-2.49** | * | **DE** | **12.33** **3.82** | **1.65** **105.80** | **45** | **67** | **16** |
| | 6.57 | 3.07 | -2.02 | * | DE | 4.92 3.05 | 2.44 85.63 | 27 | 51 | 21 |
| *Ammonium nitrate* | **6.99** | **1.96** | **-3.74** | *** | **SE** | **8.54** | **8.96** | **14** | **67** | **13** |
| | 2.03 | 0.47 | -3.27 | ** | SE | 2.44 | 8.59 | 15 | 69 | 17 |
| *V-Ni bearing* | **4.23** | **1.84** | **-3.11** | ** | **SE** | **5.66** | **10.59** | **11** | **61** | **19** |
| | 0.79 | 0.44 | -2.96 | ** | L | 1.13 | 11.94 | 8 | 57 | 25 |
| *Industrial/Metallurgy (BCN)* | **1.64** | **0.71** | **-3.43** | *** | **DE** | **1.89** **0.0015** | **7.41** **-2.06** | **29** | **56** | **11** |
| *Mineral* | **4.61** | **3.01** | **-1.87** | + | **L** | **6.53** | **16.19** | **12** | **64** | **46** |
| | 3.46 | 2.32 | -1.71 | + | L | 3.34 | 27.46 | 5 | 46 | 34 |
| *Marine* | **5.67** | **4.91** | **-1.40** | | **L** | **6.45** | **47.09** | **1** | **19** | **14** |
| | 1.53 | 1.89 | 0.62 | | L | 1.71 | -107.83 | 0 | -10 | 24 |
| *Industrial/Traffic (MSY)* | 2.08 | 1.01 | -3.11 | ** | L | 2.18 | 12.94 | 7 | 54 | 11 |
| *Aged Organics (MSY)* | 3.14 | 3.26 | 1.25 | | L | 3.34 | -51.05 | 1 | -22 | 17 |
| *Traffic (BCN)* | **8.21** | **3.33** | **-2.65** | ** | **SE** | **9.87** | **8.82** | **15** | **68** | **21** |
| *Road/work resuspension (BCN)* | **7.33** | **0.89** | **-2.80** | ** | **SE** | **10.52** | **6.15** | **27** | **80** | **41** |















**Figure Captions:**

**Figure 1**: Location of the Barcelona (BCN) and Montseny (MSY) measuring stations. Red full circle highlights the location of the BCN measuring station before 2009. Green full circle highlights the new location of the BCN (from 2009) and MSY measuring stations.

**Figure 2**: Mann-Kendall (a) and Multi-exponential (b) fits of $PM_{2.5}$ trends at MSY station for the periods 2002-2010 (as in Cusack et al., 2012), 2004 – 2014 (this work), and 2002 – 2014 (largest period so far available). TZ_MK: Mann-Kendall trend (%/yr); S_MK: significance of Mann-Kendall trend; $a_n$: multiplicative constant of exponential function ($\mu g/m^3$); $\tau_n$: characteristic time of exponential function (yr); NL: non-linearity (%); TR: total reduction (%); RC: residual component (%). Significance of the trends following the Mann-Kendall test: *** (p-value < 0.001), ** (p-value < 0.01), * (p-value < 0.05), + (p-value < 0.1).

**Figure 3**: Mann-Kendall (MK) and Multi-exponential (ME) trends for chemical species at BCN in $PM_{10}$. Measured concentration (green line); Multi-exponential trend (red line); Multi-exponential residuals (blue line); Mann-Kendall trend (black line); Mann-Kendall residuals (grey line). Trend type: linear (L), single-exponential (SE), double exponential (DE).

**Figure 4**: Mann-Kendall (MK) and Multi-exponential (ME) trends for chemical species at MSY in $PM_{10}$. Measured concentration (green line); Multi-exponential trend (red line); Multi-exponential residuals (blue line); Mann-Kendall trend (black line); Mann-Kendall residuals (grey line). Trend type: linear (L), single-exponential (SE), double exponential (DE).

**Figure 5**: Source contributions from PMF model in $PM_{10}$ at Montseny (MSY) and Barcelona (BCN). Mean values during 2004-2014. Values reported are: ***Source; $\mu g/m^3$; %.***

**Figure 6:** Mann-Kendall and Multi-exponential trends for source contributions in $PM_{10}$ at BCN. Measured concentration (green line); Multi-exponential trend (red line); Multi-exponential residuals (blue line); Mann-Kendall trend (black line); Mann-Kendall residuals (grey line). Trend type: linear (L), single-exponential (SE), double exponential (DE). Highlighted with yellow colour the source contributions at BCN from *Mineral*, *Traffic* and *Road/work resuspension* were excluded from the trend discussion.

**Figure 7**: Mann-Kendall and Multi-exponential trends for source contributions in $PM_{10}$ at MSY. Measured concentration (green line); Multi-exponential trend (red line); Multi-exponential residuals (blue line); Mann-Kendall trend (black line); Mann-Kendall residuals (grey line). Trend type: linear (L), single-exponential (SE), double exponential (DE).

**Figure 8**: NASA $NO_2$ OMI level 3 plotted using the Giovanni online data system, developed and maintained by the NASA GES DISC.

**Figure 9**: Annual (2004–2014) energy consumption for Spain (normalized to year 2004). Data from the Spanish Ministry of Industry (MINETUR, 2013).

**Figure 10**: Spanish national emission of $SO_2$ and $NO_X$ (normalized to year 2004).



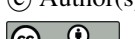

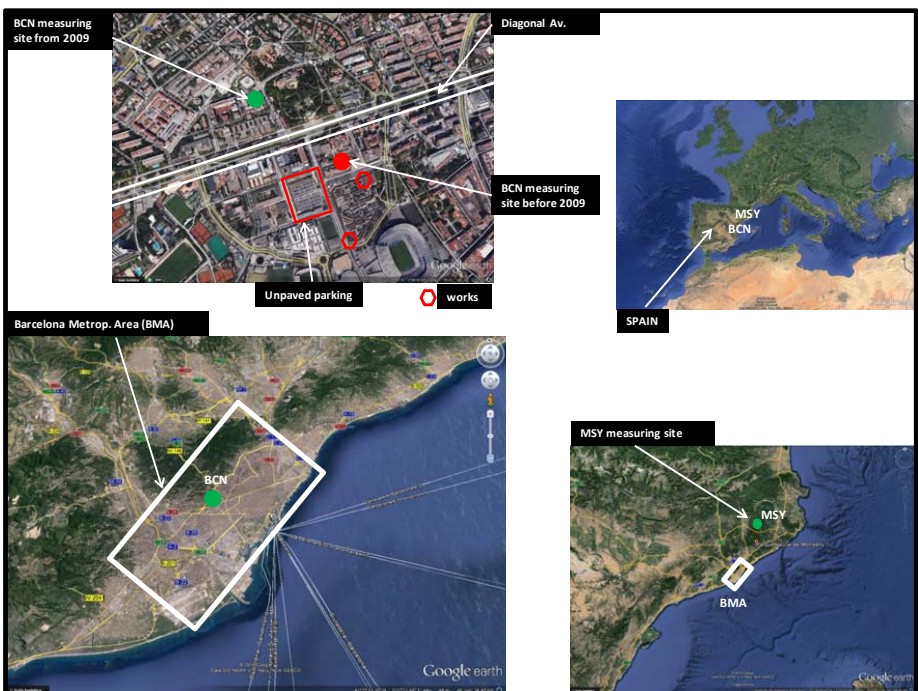

**Figure 1**





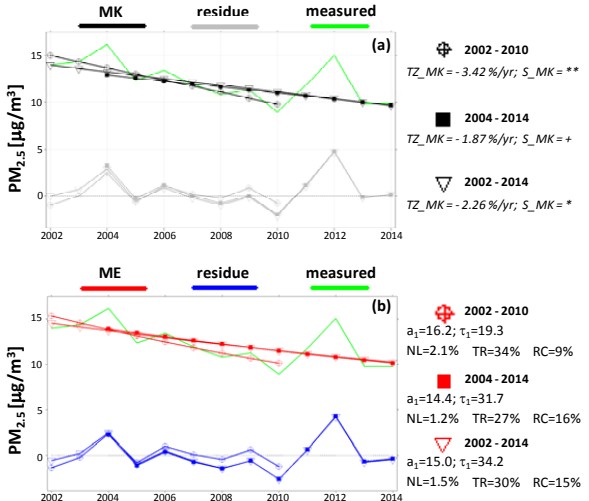


**Figure 2**




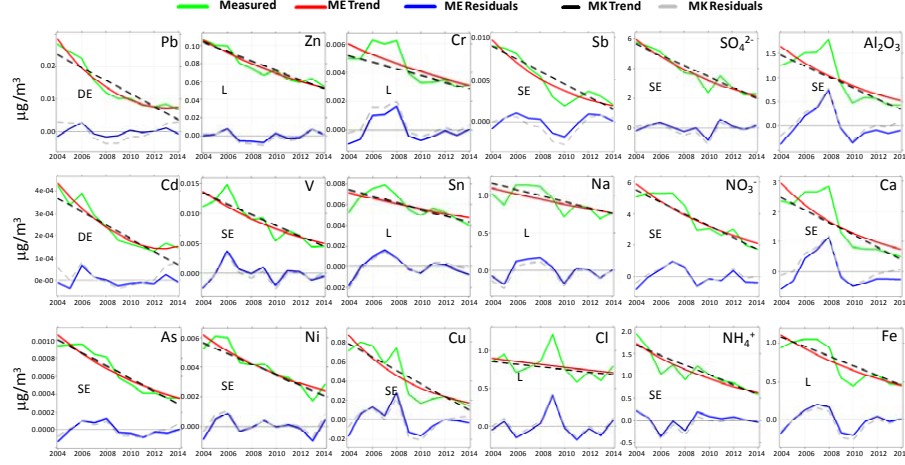


**Figure 3**








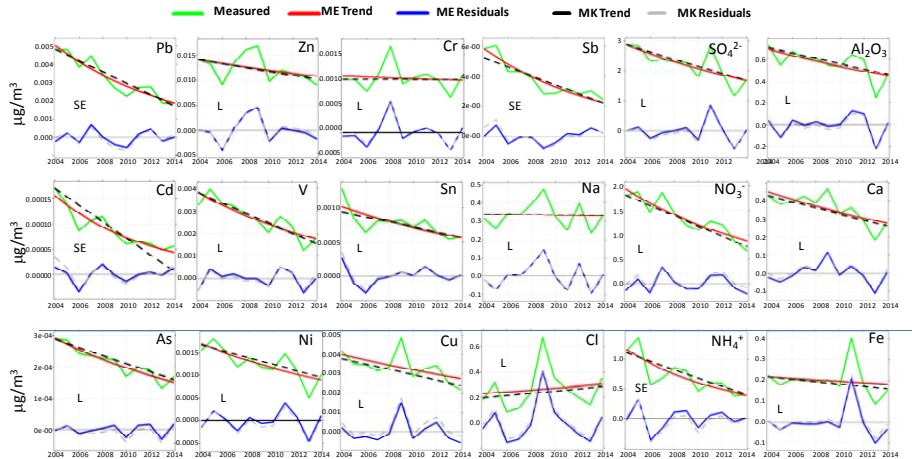


**Figure 4**




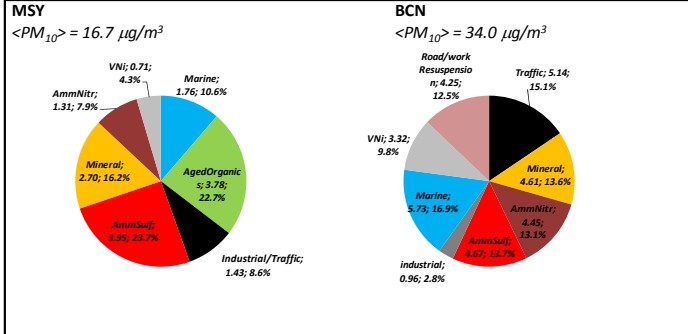


**Figure 5**










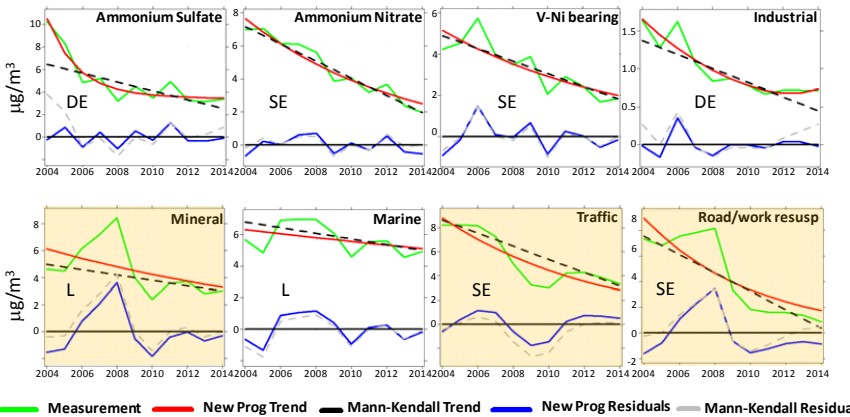

━━ Measurement   ━━ New Prog Trend   ━━ Mann-Kendall Trend   ━━ New Prog Residuals   ┄┄ Mann-Kendall Residuals
**Figure 6**




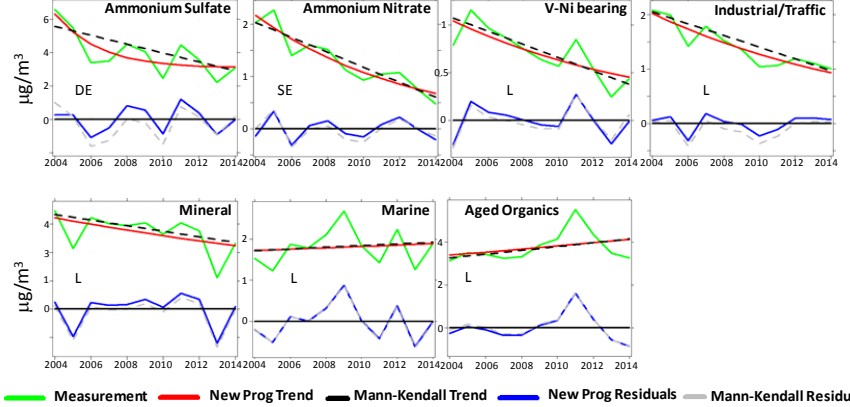

━━ Measurement   ━━ New Prog Trend   ━━ Mann-Kendall Trend   ━━ New Prog Residuals   ┄┄ Mann-Kendall Residuals
**Figure 7**



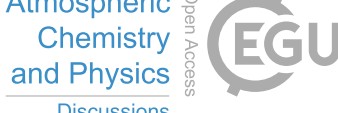
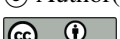


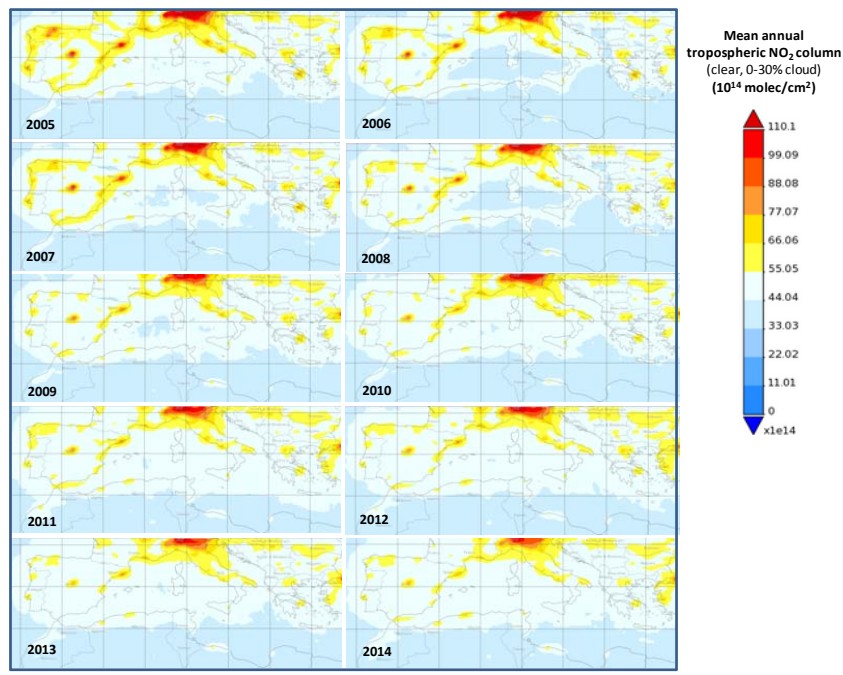


**Figure 8**














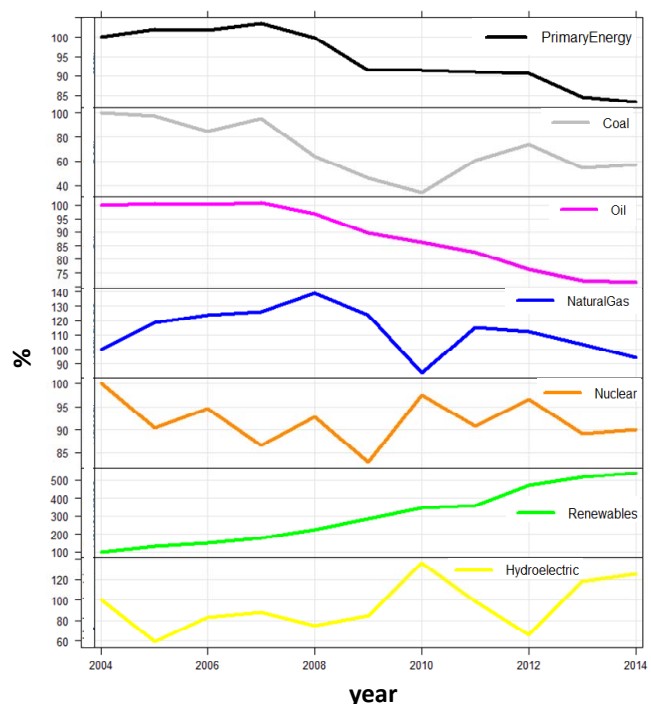


**Figure 9**



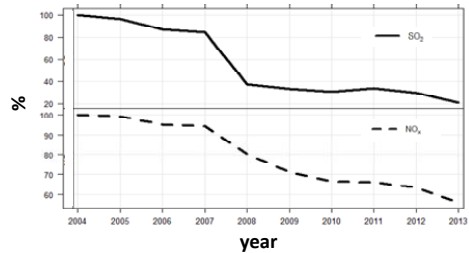


**Figure 10**

