# Peer review of "Trends analysis of PM source contributions and chemical tracers in NE Spain during 2004 - 2014"

_Atmospheric Chemistry and Physics, 2016_

## Referee Comment (RC1) · Anonymous Referee #1 · 20 Apr 2016

Journal: ACP Title: Trends analysis of PM source contributions and chemical tracers in NE Spain during 2004 - 2014: A multi-exponential approach Author(s): Pandolfi M. et al. MS No.: acp-2016-33 MS Type: Research Article

Anonymous Referee #1 Comments.

General Comments.

In this paper a trend analysis of PMx concentrations recorded at two different sites in NE Spain during the period 2004-2014 has been performed. PM10 and PM2.5 chemical composition and PM10 source contributions have also been evaluated with the aim to obtain more accurately interpretations of the trends as well as of the effectiveness of

the pollution control measures implemented in this period by the administrations. Two different methodologies have been used. Namely, the Mann-Kendall test (MK) and the Multi-Exponential fit (ME). In brief, I think that the present work shows some significant and novel contributions to the global scientific community in relation with trend analysis of atmospheric pollutants issues. I think that the availability of time series of PM chemical composition and of estimations of PM source contributions from receptor models, such as PMF, is nowadays a key factor for establishing reliable source-receptor relationships and obtaining robust results from trend analysis and even epidemiological analysis. However, some significant changes must be performed to clarify the usefulness of the different methodologies employed, to reduce the excess of information provided in the manuscript that makes the reading very hard, and to justify the behaviour observed of the trends of some of the time series analyzed.

Specific Comments.

In spite of the fact that the main results of the work (the pollution control measures effectively produced a reduction of the contributions from some anthropogenic sources in such a way that the PMx levels decreased at the urban-background and the regional background sites) are rather consistent and well justified, it is not clear the advantage of using simultaneously both trend analysis approaches, MK and ME.

In general when the ME showed a linear fit, the MK also showed a statistically significant linear trend, but sometimes it is not statistically significant, as in the case of Zn and Na in PM10 for the Montseny site. How should we interpret these different behaviours?. In the case that the MK showed a highly significant linear reduction trend and the ME showed a double exponential fit, Cd in PM10 for the Barcelona site, what result must prevail, the MK or the ME one?. Please, try to clarify the best way to take advantage of using both methodologies.

Why did you decide to work with annual mean values instead of monthly mean values?. It strongly reduced the number of data for the trend analysis.

In my opinion, the MK results can be easily interpreted by the potential readers. I mean, the test provides a statistically significant downward or upward trends for a given confidence level and the value of an estimator of the trend (%variation/year). If the test provides a non-statistically significant trend, then it must be interpreted as the absence of trend. In the case of the ME results it is not clear the meaning of linear, single-exponential and double-exponential fit in relation with the trends of the pollutants. For the double-exponential fit cases, the values of T1 and T2 are sometimes quite different, even positive and negative. Despite some explanations of these values are included in the text, the interpretation remains somewhat obscure. Some more information should be included in the 2.5 section about the interpretation of the equations and coefficients representing linear, single-exponential and double-exponential fit. Is the multi-exponential fit statistically significant in all the cases?. I suppose that the fit parameters an and Tn are estimated by the program with statistically significance for a given confidence level. As in the case of the coefficients of a multilineal regression analysis. Is not it?. Please, try to clarify all these questions in the revised version of the manuscript.

Otherwise, the ME and MK methodologies provides for any case a lot of information. I mean, the trend estimation, p-value, type of fit,. . . These results are summarized in tables 1-5. However, the authors decided to include most of this information again in the text. As a consequence the reading is very hard and confusing. For the MK results it is not necessary to include the magnitude of the trend and the degree of statistical significance in all the cases. Please, try to highlight in the manuscript only the most important information and refer to the tables for details.

I would like to underline one important fact. Sometimes it can be read in the text statements as "non-statistically significant decreasing trend (-1.25 %/year" (page 10, lines 297-298). I completely disagree with this. If the test provides a non-statistically significant trend result, you cannot assure the existence of a trend, neither downward nor upward. That is the aim of performing statistical tests. Accepting or rejecting null

hypothesis with statistical significance. Hence, you can neither talk about increasing or decreasing trend nor show the value of the estimator of the trend in the non-statistically significant trend cases. I believe that you must rewrite the manuscript and the tables, excluding the values of the estimator of the trends for non-statistically significant cases.

The reason why you decide not to use the data of mineral matter and road traffic emissions from the Barcelona site before 2009, has been repeatedly mentioned across the manuscript.

The statement that "2002‐2014 represented the largest period of gravimetric PM2.5 measurements available at MSY station" has been also repeated unnecessarily.

In page 10 (lines 304-310) and page 13 (lines 402-403) you mentioned results from subsequent sections. I think it should be avoided for clarity.

In section 3.2 you attributed the differences observed in the magnitude of the trends for different time periods to meteorology variability. In section 3.3 (lines 509-511) you stated that "It is probable that variations in meteorological conditions from one year to another (i.e. intensity and frequency of Saharan dust outbreaks) might also explain the observed trend of mineral tracers at regional level". In section 4.1 (lines 656-658) you also declared that "This decreasing trend could be due to a possible decrease of the emissions of anthropogenic mineral species from specific sources such as cement and concrete production and construction works". These comments are highly speculative due to the fact that neither meteorological variables, nor information on "intensity and frequency of Saharan dust outbreaks", nor information on "cement and concrete production and construction works" have been analyzed to support them. In my opinion it should be mandatory to carry out an analysis of this kind of data to confirm these hypotheses.

Some other minor comments and suggestions:

Did you achieve a PMF source contribution study with the PM2.5 data base?. It should

be interesting to compare the results of the trend analysis for PM10 and PM2.5 source contributions.

Page 6, lines 188-189. "(End user's guide to multilinear engine applications from Pentti Paatero)". What do you mean?. Is that a reference?.

Page 6, line 190. You declared that rotational ambiguity of the PMF solution was handled by means of the Fpeak parameter. However, some better tests to estimate rotational uncertainty than Fpeak are now available in the latest version of EPA PMF (V.5.0) such as the base model displacement error estimation and other rotational tools. Have you checked these new options?.

Page 7. Section 2.4. The description of the Mann-Kendall test is very short. Some more information should be included.

Page 8. Lines 230-232. What does Cbeg and Cend mean?.

Page 9. Lines 259-261. The description of the section was made before in the previous section. This paragraph can be omitted.

Page 9. Lines 263-264. "Note that the recommended annual data coverage for trend studies is typically 75%". Where does it come from?. Can you include the source in the Reference section?.

Page 9. Lines 276-277. The results of the comparison between simultaneous PMx chemical speciated data collected at both BCN measurement sites can be showed as Supplementary Information.

Page 10. Line 317. What is the definition of the Residual Component (RC)?.How did you computed it?. This information should be included in Section 2.

Page 20. Line 633. Querol et al. 2007, is not included in the Reference section.

Technical corrections/Typing errors.
* * *
Figure 2 has very low quality. It is very hard to distinguish among the different symbols. The grey lines and symbols are very diffuse. This is also true for figures 3, 4, 6 and 7.

Table 4 is unnecessary. This information is showed in Figure 5.

---

## Referee Comment (RC2) · Anonymous Referee #3 · 27 Jun 2016

The main problem with this manuscript is that it reads like a technical report rather than a scientific publication. Therefore, although the analysis made in this paper may be scientifically sound, the outline of the manuscript requires some fundamental revisions before I can recommend accepting this paper for publication. My detailed comments in this regard are given below.

Major comments

The authors should define clearly the scientific goals/aims of this paper. Currently, the last paragraph of section 1 merely lists what has been done in the paper without specifying what the authors aim to solve or find out in doing all this analysis.

[Figure]

Section 3, and especially 3.3, like a huge number of trend values which all can be found in the tables. This not only makes the text extremely unappealing to read, but also the most important findings of this analysis remain hidden behind these numbers. I strongly recommend the authors i) to shorten this section considerably, ii) to remove most of the trend values from the actual text, and iii) to bring up more explicitly the most important findings.

The paragraph on line 79-97 does not fit to the introduction of a scientific paper. Some of it could be part of the methods section, if needed. Even then, any citations to Wikipedia are highly questionable.

Section 5 is currently a single long paragraph. I would recommend the authors to organize this section better and perhaps to put different types of conclusions in separate pagraphs.

Minor/technical comments

Various sections should be called sections, not paragraphs in the text.

Ammonium sulfate and ammonium nitrate sound a bit strange names for sources, as they are compounds that originate from a number of sources as a result of atmospheric processing.

Line 156: please specify EUSAAR protocol.

The language of the paper requires some improvements here and there. Some examples: - there is something wrong the sentence on lines 76-78 - line 133: . . .will be discussed later - line 250: . . .(not shown), we concluded. . . - line 270: . . .what was observed. . . - there is something wrong the sentence on lines 280-284

---

## Author Comment (AC1) · 1 Sep 2016

We thank the Reviewer#1. With his/her comments the quality and readability of the manuscript have been strongly improved.

The GREEN colour was used to highlight changes in the revised manuscript rose from the comments from Referee#1.

**General Comments.**

*In this paper a trend analysis of PMx concentrations recorded at two different sites in NE Spain during the period 2004-2014 has been performed. PM10 and PM2.5 chemical composition and PM10 source contributions have also been evaluated with the aim to obtain more accurately interpretations of the trends as well as of the effectiveness of the pollution control measures implemented in this period by the administrations. Two different methodologies have been used. Namely, the Mann-Kendall test (MK) and the Multi-Exponential fit (ME). In brief, I think that the present work shows some significant and novel contributions to the global scientific community in relation with trend analysis of atmospheric pollutants issues. I think that the availability of time series of PM chemical composition and of estimations of PM source contributions from receptor models, such as PMF, is nowadays a key factor for establishing reliable source-receptor relationships and obtaining robust results from trend analysis and even epidemiological analysis. However, some significant changes must be performed to clarify the usefulness of the different methodologies employed, to reduce the excess of information provided in the manuscript that makes the reading very hard, and to justify the behaviour observed of the trends of some of the time series analyzed.*

**Specific Comments.**

1) *In spite of the fact that the main results of the work (the pollution control measures effectively produced a reduction of the contributions from some anthropogenic sources in such a way that the PMx levels decreased at the urban-background and the regional background sites) are rather consistent and well justified, it is not clear the advantage of using simultaneously both trend analysis approaches, MK and ME. In general when the ME showed a linear fit, the MK also showed a statistically significant linear trend, but sometimes it is not statistically significant, as in the case of Zn and Na in PM10 for the Montseny site. How should we interpret these different behaviours?. In the case that the MK showed a highly significant linear reduction trend and the ME showed a double exponential fit, Cd in PM10 for the Barcelona site, what result must prevail, the MK or the ME one?. Please, try to clarify the best way to take advantage of using both methodologies.*

        In order to take into account the Referee's comment and to reduce the excess of information provided in the manuscript the following changes were performed:

a) In the revised version of the manuscript ME and MK tests are not presented together and or MK results or ME results are presented. Thus, when the trend is exponential (linear) the results from Mann-Kendall test (Multi-exponential test) are not presented. In this way, we highlight in the manuscript the fact that the two fits are different and that the MK test does not properly fit the data when these have actually an exponential trend. Similarly, there is no need to use the ME test when the trend is linear. This will avoid creating confusion for the readers. In fact, when the trend is exponential (single or double) the results from the MK test are always statistically significant; however data are not well fitted using MK in this case. On the other side, when the trend is linear (not exponential) we do not need the ME test and the trend can be or not statistically significant. Moreover, with this change we also highlight in the manuscript the main difference between the MK and ME fits: when a trend is exponential it means that the observed decreasing trend was not constant and gradual over time (as indicated by the MK fit).

b) We also removed from the manuscript (Figures and Tables) those species or source contributions showing no statistically significant trends In this way we further reduce excess information presented in the manuscript. Species or sources showing no trend are now commented in the manuscript but are removed from Tables and Figures.

2) *Why did you decide to work with annual mean values instead of monthly mean values?. It strongly reduced the number of data for the trend analysis.*

We show in the manuscript that despite the rather low annual data coverage of filter data, the trends of the annual means can be reasonably studied. However, the use of monthly means could introduce undesired noise given that filter collection is not always evenly distributed along the year (i.e. because of intensive campaigns, technical problems,.....). Given that the basic information (trend magnitude and statistically significance) provided using annual or monthly means is similar we would like to use the annual means in our manuscript.

3) *In my opinion, the MK results can be easily interpreted by the potential readers. I mean, the test provides a statistically significant downward or upward trends for a given confidence level and the value of an estimator of the trend (%variation/year). If the test provides a non-statistically significant trend, then it must be interpreted as the absence of trend. In the case of the ME results it is not clear the meaning of linear, single exponential and double-exponential fit in relation with the trends of the pollutants. For the double-exponential fit cases, the values of T1 and T2 are sometimes quite different, even positive and negative. Despite some explanations of these values are included in the text, the interpretation remains somewhat obscure. Some more information should be included in the 2.5 section about the interpretation of the equations and coefficients representing linear, single-exponential and double-exponential fit. Is the multi-exponential fit statistically significant in all the cases?. I suppose that the fit parameters an and Tn are estimated by the program with statistically significance for a given confidence level. As in the case of the coefficients of a multilineal regression analysis. Is not it?. Please, try to clarify all these questions in the revised version of the manuscript.*

We agree with the reviewer that the statistical significance of the exponential fits is missing in the manuscript. In the revised version of the manuscript p-values are provided for both linear (Mann-Kendall) and exponential fits. Tables 2,3 and 5 and text were opportunely modified. Moreover, the following text was added to Paragraph 2.5 in order to help the reader to interpret the results from the exponential fit.

"The main difference between linear and exponential fit is that in the latter case the trend is not gradual and constant over time. For an exponential trend the absolute [$\mu g/m^3$] reduction per year decreases with time being the highest at the beginning of the period. Conversely, for a linear fit the absolute reduction is constant over time. For an exponential fit, the lower the characteristic time $\tau$ the more rapidly the considered quantity vanish. Deviations from single exponential fit can be taken into account introducing more exponential terms. In this work for example two exponential terms were sometime used. In this case, two characteristic times are calculated by the software. If the decrease of the considered quantity is very sharp at the beginning of the period (more than exponential) than both $\tau_1$ and $\tau_2$ are positive. Conversely, an exponential term with negative $\tau$ takes into account

for possible increases of the quantity at the end of the period. Both $\tau_n$ and $a_n$ are calculated by the program by means of the least square method minimizing the residue $\omega$ and the statistically significance of the exponential fit is provided by means of the p-value."

4) *Otherwise, the ME and MK methodologies provides for any case a lot of information. I mean, the trend estimation, p-value, type of fit… These results are summarized in tables 1-5. However, the authors decided to include most of this information again in the text. As a consequence the reading is very hard and confusing. For the MK results it is not necessary to include the magnitude of the trend and the degree of statistical significance in all the cases. Please, try to highlight in the manuscript only the most important information and refer to the tables for details.*

Following the reviewer comment, in the revised manuscript the text was shortened as much as possible in order to avoid repeated information. Moreover, as already stated in the answer to the *specific comment #1*, or MK or ME results (not both) are presented in the revised manuscript, thus further helping in making the text more readable.

5) *I would like to underline one important fact. Sometimes it can be read in the text statements as "non-statistically significant decreasing trend (-1.25 %/year" (page 10, lines 297-298). I completely disagree with this. If the test provides a non-statistically significant trend result, you cannot assure the existence of a trend, neither downward nor upward. That is the aim of performing statistical tests. Accepting or rejecting null hypothesis with statistical significance. Hence, you can neither talk about increasing or decreasing trend nor show the value of the estimator of the trend in the non-statistically significant trend cases. I believe that you must rewrite the manuscript and the tables, excluding the values of the estimator of the trends for non-statistically significant cases.*

We agree with the reviewer. Consequently, we removed from the manuscript (Figures and Tables) those species or source contributions showing no statistically significant trends. In this way we further reduce excess information presented in the manuscript. Species or sources showing no trend are now commented briefly in the manuscript but are removed from Tables and Figures.

6) *The reason why you decide not to use the data of mineral matter and road traffic emissions from the Barcelona site before 2009, has been repeatedly mentioned across the manuscript.*

*And*

7) *The statement that "2002-2014 represented the largest period of gravimetric PM2.5 measurements available at MSY station" has been also repeated unnecessarily.*

Repetitions are avoided as much as possible in the revised manuscript.

8) *In page 10 (lines 304-310) and page 13 (lines 402-403) you mentioned results from subsequent sections. I think it should be avoided for clarity.*

The sentence at page 10, lines 304-310: "We will show later (*Paragraph 4.1*) that despite the fact that the gravimetric concentrations of PM$_{10}$ at MSY only slightly decreased monotonically with time (-0.47 %/yr with p>0.1; cf. Table 1), the contributions from specific PM$_{10}$ pollutant sources from PMF model related with anthropogenic activities showed non linear (i.e. exponential) statistically significant decreasing trends. For example the *Ammonium sulfate* source contribution to PM$_{10}$ at MSY decreased at the rate of -2.02 %/yr with p<0.05 310 from MK test and double exponential fit of the data was needed; cf. Table 5)."

Was replaced with:

"Given that the trends of the considered PMx fractions were linear at both sites (NL<10%), only results from MK test were reported in Table 1. However, we will show later (Paragraph 4.1) that the contributions from specific PM$_{10}$ pollutant sources from PMF model, mainly those related with anthropogenic activities, showed non-linear (i.e. exponential) decreasing trends, thus mirroring the different effectiveness of the mitigation strategies depending on the source of pollutants considered."

Moreover, the sentence at page 13, lines 402-403: "Interestingly, as shown later, the PM10 Industrial/metallurgy source contribution at BCN also showed a DE decreasing trend", was removed from the text.

9) *In section 3.2 you attributed the differences observed in the magnitude of the trends for different time periods to meteorology variability. In section 3.3 (lines 509-511) you stated that "It is probable that variations in meteorological conditions from one year to another (i.e. intensity and frequency of Saharan dust outbreaks) might also explain the observed trend of mineral tracers at regional level". In section 4.1 (lines 656-658) you also declared that "This decreasing trend could be due to a possible decrease of the emissions of anthropogenic mineral species from specific sources such as cement and concrete production and construction works". These comments are highly speculative due to the fact that neither meteorological variables, nor information on "intensity and frequency of Saharan dust outbreaks", nor information on "cement and concrete production and construction works" have been analyzed to support them. In my opinion it should be mandatory to carry out an analysis of this kind of data to confirm these hypotheses.*

In section 3.2 we compared the trends of PM$_{2.5}$ mass concentrations over different periods with the main aim of: a) being consistent with what already published at MSY station by Cusack et al. (2011) where 9 yr of data (2002-2010) were used, and b) to study the differences in the trends over short periods (9 yr to 13 yr). The fact that the trends were statistically significant over different periods confirms the effectiveness of the measures taken to improve air quality.

To take into account the reviewer comment, the following sentences were removed from Pragraph 3.2:

"However, the difference observed in the magnitude of the trends during 2004-2014 compared to the results provided by Cusack et al. (2012) suggested that meteorology (in this case a large increase in 2012; cf. Figure 2), changing from year to year, also determined the degree of comparability of trends observed over different periods."

and

"Thus, over relatively short periods (9 -11 yr), the effects of just one meteorologically different year were clearly visible.".

And the following sentence was added at the end of Pragraph 3.2:

"However, it should be noted that the statistical significance of the trends observed for the larger periods was lower compared to Cusack et al. (2012). The difference observed in the magnitude of the trends during 2004-2014 compared to the results provided by Cusack et al. (2012) was mainly due to the increase of $PM_{2.5}$ mass concentration in 2012 (cf. Figure 2). Chemical $PM_{2.5}$ speciated data revealed that this increase was partly driven by organic matter showing a mean annual concentration in 2012 higher by around 20% compared to the 2004-2014 average."

Moreover, in Paragraph 3.3 we reported that the total reduction observed for the concentrations of mineral species was higher in $PM_{2.5}$ mass fraction compared to $PM_{10}$, thus suggesting an anthropogenic contribution to the mass of mineral matter in $PM_{2.5}$ and that this anthropogenic contribution likely decreased with time as already observed by Cusack et al. (2012).

In order to better explains possible reasons for the trends observed for mineral matter the following sentence:

"As for Cr, Sn, Sb and Cu, the trends of mineral species (Al2O3, Ca, Fe) were studied only at MSY station. For these elements, linear (with the exception of Al2O3 in PM2.5 which was SE) and statistically significant decreasing trends (with the exception of Ca in PM2.5 with p>0.1) were detected. On average the TR were higher in the fine fraction, ranging from 50% for Ca to 66% for Al2O3, compared to PM10 (16-38% cf. Table 2) thus likely suggesting a decrease with time of the concentrations of anthropogenic mineral species from specific sources such as cement and concrete production and production works. In fact, coarse mineral matter at regional background sites is mainly of natural origin. Downward decreasing trend for mineral matter contribution in PM2.5 at MSY was also reported by Cusack et al. (2012) for the period 2002 – 2010 at the same station. It is probable that variations in meteorological conditions from one year to another (i.e. intensity and frequency of Saharan dust outbreaks) might also explain the  observed trend of mineral tracers at regional level."

Was replaced with the following sentence:

"For the mineral species (Al2O3, Ca, Fe) linear (with the exception of Al2O3 in PM2.5 which was SE) and statistically significant decreasing trends were detected at MSY. On average the TR was higher in the fine fraction, ranging from 50% for Ca to 66% for Al2O3, compared to PM10 (6-38% cf. Table 2). Downward

decreasing trend for crustal material in PM2.5 at MSY was also reported by Cusack et al. (2012) for the period 2002 – 2010 and by Querol et al. (2014) for the period 2001 – 2012. These trends were probably driven by weather conditions associated with negative NAO index (iNAO) that could be the cause for this slight reduction observed in crustal material. Pey et al. (2013) found a correlation between iNAO (calculated between June and September) and the contribution of Saharan dust to PM10 mass in NE of Spain showing that the more negative is the iNAO the lower is the dust contribution to PM. The iNAO was unusually negative during the period 2008 – 2012 (http://www.cpc.ncep.noaa.gov/products/precip/CWlink/pna/norm.nao.monthly.b5001.current.ascii) thus likely contributing to explain the observed trends of crustal elements. Moreover, negative NAO can favour the presence of fronts that can sweep the Iberian Peninsula from West to East causing higher wind and less stagnant conditions thus favouring the dispersion of pollutants. In addition, as suggested by Cusack et al (2012), it could also be hypothesised that some part of the crustal material measured at MSY is a product of the construction industry. The construction industry in Spain has been especially affected by the current economic recession, and crustal material produced by this industry may have contributed to the crustal load in PM2.5. For example, the number of home construction works in Barcelona during 2008 – 2014 (from the beginning of the economic crisis; mean number of works = 1281) reduced by around 75% compared to the period 2000 – 2007; mean number of works = 5187) (http://www.bcn.cat/estadistica/castella/dades/timm/construccio/index.htm). The fact that the total reduction calculated for mineral elements reported in Tables 2 and 3 was higher in PM2.5 compared to PM10 could corroborate this latter hypothesis."

Moreover, the following sentence in Paragraph 4.1:

"The statistically significant decreasing trend observed at MSY for the Mineral source was in agreement with what observed at the same station by Cusack et al. (2012). This decreasing trend could be due to a possible decrease of the emissions of anthropogenic mineral species from specific sources such as cement and concrete production and construction works."

Was replaced with the following sentence:

"Finally, the Mineral source contribution at MSY showed linear little significant decreasing trend (p<0.1) in agreement with what observed at the same station by Cusack et al. (2012). As already noted in Paragraph 3.3, this negative trend could be due to both a possible decrease of the emissions of finer anthropogenic mineral species from specific sources such as cement and concrete production and construction works and unusual weather conditions reducing Saharan dust contribution to PM and resuspension of dust."

**Minor Comments.**

*10) Did you achieve a PMF source contribution study with the PM2.5 data base?. It should be interesting to compare the results of the trend analysis for PM10 and PM2.5 source contributions.*

Before starting with PMF analysis on PM10 chemical speciated data, the PM10 database for Montseny station was opportunely revised. Thus, before this work, only total carbon concentration was available for the period 2004 – 2007. In order to properly apply the PMF model, the concentrations of elemental (EC) and organic (OC) carbon in PM10 were recovered from filters sampled during the period 2004 – 2007. This was not done in PM2.5. Given that the data recover implies laboratory analysis which takes rather long time we decided to apply the PMF model to the PM10 mass fraction and to present trends of chemical species for both PM mass fractions.

In order to not delay the publication of this work and with the permission of the Reviewer, we would like to present PM10 source apportionment results only.

11) *Page 6, lines 188-189. "(End user's guide to multilinear engine applications from Pentti Paatero)". What do you mean?. Is that a reference?.*

The correct reference is (Paatero, 2004). We added the following reference to the bibliography:

"Paatero P.: User's guide for positive matrix factorization programs PMF2 and PMF3, Part1: tutorial. University of Helsinki, Helsinki, Finland, 2004"

12) *Page 6, line 190. You declared that rotational ambiguity of the PMF solution was handled by means of the Fpeak parameter. However, some better tests to estimate rotational uncertainty than Fpeak are now available in the latest version of EPA PMF (V.5.0) such as the base model displacement error estimation and other rotational tools. Have you checked these new options?.*

The aim of this work was to study trends of source contributions. To do this we applied the PMF model to 11 yr of chemical speciated data. Thus, we supposed that the chemical profiles of the detected sources did not change with time. To test this hypothesis we performed a "sliding PMF" (using groups of 3 years of data: i.e. 2004-2006; 2005-2007; 2006-2008 and so on) comparing the retrieved source profiles. The results suggested that the error in detecting source profiles and contributions was lower using the whole (11 years) database. For example, the PMF was not able to clearly detect the *Industrial/Traffic* and *V-Ni bearing* sources at MSY at the end of the considered period by means of the sliding PMF. Thus, we concluded that the best option was to use the whole database as input in the PMF model even if changes in the chemical profiles cannot be excluded. However, only in this way, the trends of the contributions of some sources such as *Industrial/Traffic* and *V-Ni* agreed well with the trends of the main tracers of these sources. Thus, given the large period used for PMF analysis, the deviation from constant profiles could largely affect the displacement results and errors.

13) *Page 7. Section 2.4. The description of the Mann-Kendall test is very short. Some*

*more information should be included.*

Paragraph 2.4 was modified as it follows:

"The purpose of the Mann-Kendall (MK) test (Mann 1945, Kendall 1975, Gilbert 1987) is to statistically assess if there is a monotonic upward or downward trend of the variable of interest over time. A monotonic upward (downward) trend means that the variable consistently increases (decreases) through time. The Mann-Kendall test tests the null hypothesis H0 of no trend, i.e. the observations are randomly ordered in time, against the alternative hypothesis, H1, where there is an increasing or decreasing monotonic trend. The main advantage of the Mann-Kendall test is that data need not conform to any particular distribution and missing data are allowed. To estimate the slope of the trend the Sen's method was used (Salmi et al. 2002)."

14) *Page 8. Lines 230-232. What does Cbeg and Cend mean?.*

The following sentence was added after Equation 5:

"Where $C_{beg}$ and $C_{end}$ are respectively the first and the last points of the exponential fit."

15) *Page 9. Lines 259-261. The description of the section was made before in the previous section. This paragraph can be omitted.*

The paragraph was removed from the text

16) *Page 9. Lines 263-264. "Note that the recommended annual data coverage for trend studies is typically 75%". Where does it come from?. Can you include the source in the Reference section?.*

Recommended annual data coverage of 75% was recently set by the Task Force on Measurements and Modeling (TFMM-CLRTAP) in order to study trends of different pollutants in EU. However, given that this limit can be considered as quite arbitrary, the sentence was removed from the manuscript. Consequently, the paragraph was modified as follow:

"Annual data coverage is an important factor to take into account in order to study trends of a given parameter. The gravimetric PM measurements, from which chemical speciated data are obtained, are typically performed with rather low frequency over one year. In our case the annual data coverage of gravimetric measurements was around 20-30% at both Barcelona and Montseny. In this section we compare the trends of PM concentrations from gravimetric and real-time optical measurements (Table 1)."

17) *Page 9. Lines 276-277. The results of the comparison between simultaneous PMx*

*chemical speciated data collected at both BCN measurement sites can be showed as*

*Supplementary Information.*

The current location of the BCN measurement station is called *Palau Reial (PR)*. The location before 2009 was called *IJA*. The 1-month simultaneous filter measurements were performed at *IJA* and at a location (called *Torre Girona; TG*) which was very close to *PR* but likely more affected by the presence of trees and some

small buildings compared to *PR*. This is why we moved again from *TG* to *PR*. The ratios between the concentrations of chemical species simultaneously measured at *IJA* and *TG* were around 0.20÷0.60 (with $R^2 <$ 0.60 and with *TG* strongly underestimating *IJA*) for mineral and traffic tracers (bad correlation). The ratios were around 0.70÷0.90 (with $R^2 > 0.60$ and with *TG* slightly underestimating *IJA*) for species not related with traffic and mineral sources (good correlation). Thus, we observed rather similar concentrations between *TG* and *IJA* for those species measured at BCN and included in the manuscript. Conversely, the bad correlation observed for mineral and traffic tracers confirmed that other traffic and mineral sources were affecting *IJA* but not *TG*. The lower concentration measured at *TG* compared to *IJA* for "good" species was very probably due to the presence of small buildings around *TG*. We are confident that the ratios between *PR* and *IJA* are very close to one. In conclusion, we used the measurements performed at *TG* and *IJA* in order to certainly exclude "bad" species from the analysis. For this reason and in order to avoid confusion we would like to not present this comparison in the manuscript.

18) *Page 10. Line 317. What is the definition of the Residual Component (RC)?.How did you computed it?. This information should be included in Section 2.*

The following sentence was added to Section 2.5

"The relative contribution of residues (Residual Component: RC) is calculated as the standard deviation of the ratios between the residue values of the fit ω (cf. Eq. 1) and the main component of the fit."

19) *Page 20. Line 633. Querol et al. 2007, is not included in the Reference section.*

The following reference was added:

"Querol, X., Viana, M., Alastuey, A., Amato, F., Moreno, T., Castillo, S., Pey, J., de la Rosa, J., Artíñano, B., Salvador, P., García Dos Santos, S., Fernández-Patier, R.,  Moreno-Grau, S., Negral, L., Minguillón, M.C., Monfort, E., Gil, J.I., Inza, A., Ortega, L.A., Santamaría, J.M., Zabalza, J.: Source origin of trace elements in PM from regional background, urban and industrial sites of Spain, Atm. Env., 41, 7219-7231, 2007."

***Technical corrections/Typing errors.***

20) *Figure 2 has very low quality. It is very hard to distinguish among the different symbols. The grey lines and symbols are very diffuse. This is also true for figures 3, 4, 6 and 7.*

All Figures were re-edited

21) *Table 4 is unnecessary. This information is showed in Figure 5.*

Table 4 was removed.

[revised manuscript text omitted]

Figure 3

[Figure]

**Figure 4**

[Figure]

**Figure 5**

[Figure]

**Figure 6**

[Figure]

**Figure 7**

[Figure]

**Figure 8**

[Figure]

**Figure 9**

[Figure]

**Figure 10**

**Supporting Information**

**1) *Effect of the change of the location of the measuring station in BCN in 2009**

[Figure]

**Figure SI-1**: Trends of $PM_{10}$ concentrations from gravimetric measurements at BCN and MSY. Red rectangle highlights the decrease of $PM_{10}$ concentration at BCN due to the change of the location of the BCN measuring station in 2009.

**2) *PMF source profiles at BCN and MSY**

[Figure]

**Figure SI-2**: Chemical profiles of the PMF sources at MSY (a) and BCN (b)

Source profiles from PMF analysis at BCN and MSY. Common sources at both BCN and MSY were: *Secondary Sulfate* (secondary inorganic source traced by $SO_4^{2-}$, $NH_4^+$ with contribution from OC), *Secondary nitrate* (secondary inorganic source traced by $NO_3^-$ and $NH_4^+$), *V-Ni bearing* (traced mainly by V, Ni and $SO_4^{2-}$ it represents the direct emissions from heavy oil combustion), *mineral* (traced by typical crustal elements such as Al, Ca, Ti, Rb, Sr), *aged marine* (traced by Na and Cl mainly with contributions from $SO_4^{2-}$ and $NO_3^-$). Non common sources at MSY were: *Industrial/Traffic source* (Anthropogenic source traced by EC, OC, Cr, Cu, Zn, As, Cd, Sn, Sb and Pb includes contributions from anthropogenic sources such as traffic and metallurgic) and *Aged organics* (traced by OC and EC mainly with maxima in summer indicating a biogenic origin). Non common sources at BCN were: *Traffic* (traced by Cnm, Cr, Cu, Sb and Fe mainly and contributing 5.14 µg/m³ (15.1%)), *Road resuspension* (traced by both crustal elements, mainly Ca, and traffic tracers such as Sb, Cu and Sn and contributing 4.25 µg/m³ (12.5%)) and *Industrial* (traced by Pb, Cd, As and Zn and contributing 0.96 µg/m³ (2.8%).

**3) OC:EC ratio statistic at Montseny (MSY)**

[Figure]

**Figure SI-3**: OC and EC scatterplot (a) and frequency distribution of the OC:EC ratio (b) at Montseny (MSY) station.

Mean and median values of the OC:EC ratio at MSY were 9.1 and 7.8, respectively.

**4) PMF Barcelona: 2007-2014 (with OC and EC) vs. 2004 – 2014 (with Cnm)**

[Figure]

**Figure SI-4**: Comparison between PMF results at BCN obtained using the period 2007-2014 (separate OC and EC measurements available) and using the whole period 2004-2014 (Cnm was used in PMF).

**5) OC:EC ratio statistic at Barcelona (BCN)**

[Figure]

**Figure SI-5**: OC and EC scatterplot (a) and frequency distribution of the OC:EC ratio (b) at Barcelona (BCN) station.

---

## Author Comment (AC2) · 1 Sep 2016

**Answer to the comments from Referee#3**

We thank the Reviewer#3. With his/her comments the quality and readability of the manuscript have been strongly improved.

The **BOLD GREEN colour** was used to highlight changes in the revised manuscript rose from the comments from Referee#3.

*The main problem with this manuscript is that it reads like a technical report rather than a scientific publication. Therefore, although the analysis made in this paper may be scientifically sound, the outline of the manuscript requires some fundamental revisions before I can recommend accepting this paper for publication. My detailed comments in this regard are given below.*

**Major comments**

1) *The authors should define clearly the scientific goals/aims of this paper. Currently, the last paragraph of section 1 merely lists what has been done in the paper without specifying what the authors aim to solve or find out in doing all this analysis.*

The Section 1 was changed accordingly to the Referee's comment and the following sentences were added:

"Thanks to the aforementioned measures, there is clear evidence that the concentrations of PM in many European countries have markedly decreased during the last decades. However, in spite of the above policy efforts, a significant proportion of the urban population in Europe lives in areas exceeding the World Health Organisation (WHO) air quality (AQ) standards i.e. for $PM_{2.5}$, $PM_{10}$ and ozone (EEA, 2013, 2015).

Trend analysis of the concentration of air pollutants helps in evaluating the effectiveness of specific AQ measures depending on the pollutant considered. Examining data over time also makes it possible to predict future frequencies and/or rates of occurrence making future projections. For the abovementioned reasons, it is especially attractive the feasibility of studying the trends of the contributions to PM mass from specific pollutant sources along with the trends of the chemical tracers of these sources.

For what we are concerned in the majority of studies dealing with trend analysis, linear fits were applied for example by using Mann-Kendall or Theil-Sen methods (Theil, 1950; Sen, 1968), the latter being available for example in the Openair software (Carslaw, 2012; Carslaw and Ropkins, 2012). However, linear fit of data does not always properly represent the observed trends. As we will show, different abatement strategies and periods of implementation may change from one pollutant to another thus leading to different trends for different pollutants, even over the same period. Thus, non-linear fit of the data may be at times strongly recommended.

The main aim of this work was to study the trends of source contributions to PM10 and specific chemical species in both PM10 and PM2.5 using both the consensus methodology for linear fit of the data (Mann-Kendall) and a

non-linear approach. The data of Spanish national emissions and energy consumption are also evaluated to interpret the observed trends. Understanding past trends may be relevant for devising new strategies for air pollution abatement. PM chemical speciated data collected from 2004 to 2014 at regional (Montseny; NE Spain) and urban (Barcelona, NE Spain) sites were used with this aim. The selected period allowed for trend analysis at these twin stations over a common period. The Positive Matrix Factorization (PMF) model was used to apportion ambient PM10 concentrations into pollutant sources. The PMF model, as other Receptor Models (RM), is widely used being a powerful tool to help policy makers to design more targeted approaches to protecting public health. Thus, the novelty of this study lies mainly in a) the opportunity to study the trends of pollutant source contributions from PMF model at two twin stations representative of the urban and regional environments in the Western Mediterranean, and, b) in the use of a novel non-linear approach for trend studies."

2) *Section 3, and especially 3.3, like a huge number of trend values which all can be found in the tables. This not only makes the text extremely unappealing to read, but also the most important findings of this analysis remain hidden behind these numbers. I strongly recommend the authors i) to shorten this section considerably, ii) to remove most of the trend values from the actual text, and iii) to bring up more explicitly the most important findings.*

In order to take into account the Referee's comment and to reduce the excess of information provided in the manuscript the following changes were performed:

a) In the revised version of the manuscript ME and MK tests are not presented together and or MK results or ME results are presented. Thus, when the trend is exponential (linear) the results from Mann-Kendall test (Multi-exponential test) are not presented. In this way, we highlight in the manuscript the fact that the two fits are different and that the MK test does not properly fit the data when these have actually an exponential trend. Similarly, there is no need to use the ME test when the trend is linear. This will avoid creating confusion for the readers. In fact, when the trend is exponential (single or double) the results from the MK test are always statistically significant; however data are not well fitted using MK in this case. On the other side, when the trend is linear (not exponential) we do not need the ME test and the trend can be or not statistically significant. Moreover, with this change we also highlight in the manuscript the main difference between the MK and ME fits: when a trend is exponential it means that the observed decreasing trend was not constant and gradual over time (as indicated by the MK fit).
b) We also removed from the manuscript (Figures and Tables) those species or source contributions showing no statistically significant trends In this way we further reduce excess information presented in the manuscript. Species or sources showing no trend are now commented in the manuscript but are removed from Tables and Figures.
c) The data of Spanish national emissions and energy consumption are also evaluated to interpret the observed trends.

3) *The paragraph on line 79-97 does not fit to the introduction of a scientific paper. Some of it could be part of the methods section, if needed. Even then, any citations to Wikipedia are highly questionable.*

This Paragraph was removed from Section 1 and moved to section 2.5:

"**2.5 Multi-exponential (ME) fit**

A Program aiming at studying trends of time series of air pollution in the multi-exponential form was developed within the The Task Force on Measurements and Modelling (TFMM) by the Meteorological Synthesizing Centre – East (MSC-E; http://www.msceast.org/) group (Shatalov et al., 2015). The TFMM together with the Task Force on Emission Inventories and Projections (TFEIP), the Task Force on Integrated Assessment Modelling (TFIAM), and Task Force on Hemispheric Transport of Air Pollution (TFHTAP) provide a fora for discussion and scientific exchange in support of the EMEP (European Monitoring and Evaluation Programme; http://www.emep.int/)

work plan which is a scientifically based and policy driven programme under the Convention on Long-range Transboundary Air Pollution (CLRTAP; http://www.unece.org/env/lrtap/lrtap_h1.html) promoting the international co-operation to solve transboundary air pollution problems. The TFMM was established in 2000 to evaluate measurements and modeling and to further develop working methods and tools. In this contest, five EMEP Centers are undertaking efforts in support of the EMEP work plan, namely the MSC-E, the Centre on Emission Inventories and Projections (CEIP; http://www.ceip.at/), the Chemical Coordinating Centre (CCC; http://www.nilu.no/projects/ccc/), the Meteorological Synthesizing Centre – West (MSC-W; http://emep.int/mscw/index_mscw.html), and the Centre for Integrated Assessment Modelling (CIAM; http://www.iiasa.ac.at/~rains/ciam.html). In 2014, the TFMM initiated a dedicated exercise to assess the efficiency of air pollution mitigation strategies over the past 20 years to assess the benefit of the CLRTAP main policy instrument. Within this exercise a software was made available by EMEP/MSC-E Center aiming at studying non-linear trends."

*4) Section 5 is currently a single long paragraph. I would recommend the authors to organize this section better and perhaps to put different types of conclusions in separate pagraphs.*

*Section 5 was opportunely reorganized in order to highlight the main finding of this work:*

*"5.0 Conclusions*

*PM chemical speciated data collected at two twin stations in NE of Spain (Barcelona: urban background station and Montseny: regional background station) during 2004 – 2014 were used to study trends of source contributions from PMF analysis and of chemical species concentrations. Despite the fact the trends of different PM fractions (PM2.5 and PM10) were linear during the period under study, the trends of specific chemical elements and source contributions were exponential demonstrating the different effectiveness and time of implementation of different reduction strategies on specific pollutant sources. Statistically significant exponential trends (p < 0.01 or 0.001) were mainly observed for the industrial tracers (Pb, Cd, As) in both PM10 and PM2.5 and at both sites. The concentrations of V and Ni showed exponential trends in BCN and linear trends at MSY likely because of the higher distance of the MSY station to the sources of V and Ni (shipping and, before 2008, energy production) compared to BCN. Traffic tracers at MSY (Sn, Cu) showed very similar linear decreasing trends with higher magnitude of the trends in the fine (PM2.5) fractions compared to PM10 likely because of possible sources of coarser Sn and Cu reducing the magnitude of the trends in the PM10 mass fraction. Sb at MSY showed marked exponential decreasing trends compared to other traffic tracers (Cu and Sn) which could be explained by a aprogressive reduction of Sb content in vehicle brakes. Secondary inorganic aerosols (SO42-, NO3- and NH4+) also showed marked decreasing trends (both linear and exponential) in both fractions and at both sites. However, in general the magnitude of the trends for these species and their statistical significance were higher at BCN compared to MSY.*

*The PM10 source contributions that showed statistically significant downward trends at both Barcelona (BCN; UB) and Montseny (MSY; RB) were from Ammonium sulfate, Ammonium nitrate, and V-Ni bearing sources. For these source contributions the decreasing trends were exponential indicating that the trends were not gradual and consistent over time and that the effectiveness of the control measures for these pollutants was stronger at the beginning of the period under study (2004-2009 approximately) compared to the end of the period (Figs. 3 and 4). Statistically significant decreasing trends were observed for the Industrial/Traffic and Mineral source (at MSY; mixed road traffic and metallurgy) and the Industrial/metallurgy source at BCN. These sources were mostly linked with anthropogenic activities and the observed decreasing trends confirmed the effectiveness of pollution control measures implemented at EU or regional/local levels. The economic crisis which started in 2008 in Spain also contributed to the observed trends. Conversely, the contributions from sources mostly linked with natural processes such as Aged Marine (at both BCN and MSY) and Aged Organic (at MSY) did not show*

*statistically significant trends. The general trends observed for the calculated PMF source contributions well reflected the trends observed for the chemical tracers of these pollutant sources. The decrease in the Ammonium sulfate source contribution was mainly attributed to the EC Directive on Large Combustion Plants implemented from 2008 in Spain, resulting in the application of fuels gas desulfurization (FGD) systems in a number of large facilities. Moreover, according to the 2008 Regional AQ Plan, the use of heavy oils and petroleum coke for power generation was forbidden around Barcelona from 2008 in favour of natural gas. As a consequence, a decrease of the contributions from the V-Ni bearing source at both sites was also observed. The decrease observed for the contribution of the Ammonium Nitrate source was mainly due to the reduction in ambient NOx concentrations. In Spain a general decrease of the concentrations of NO2 at regional level was observed and it was mainly related with the lower energy consumption related with the financial crisis. The decrease of nitrates concentrations and Ammonium nitrate source contributions around Barcelona was also attributed to the decrease of NOx emissions from the five power generation plants around the city. Moreover, a Regional AQ Plan implementing the SCRT (continuously regenerating PM traps with selective catalytic reduction for NO2) and the hybridization and shift to natural gas engines of the Barcelona´s bus fleet may have had also an influence in NOx ambient concentrations. The Industrial/Metallurgy source contribution at BCN decreased exponentially reflecting the exponential trends observed for the main tracers of this pollutant source (Pb, Cd and As). The implementation of IPPC (Integrated Pollution Prevention and Control) Directives together with a decrease in the emissions from industrial production (smelters) at a regional scale around Barcelona explained the observed trends. Overall, the magnitude of the decreasing trends of the contributions of the pollutant sources were higher at BCN compared to MSY likely because of the proximity of the BCN measurement site to anthropogenic pollutant sources compared to the MSY site. The results presented in this work clearly confirm the beneficial effect of the AQ measures taken in recent years in Europe. However, the WHO limit values of specific pollutants, PM10 and PM2.5 among these, are still exceeded especially at urban level and industrial hotspots. To meet the WHO guide levels important actions are still required for the next decade and the interpretation of past air quality trends may yield relevant outcomes for planning further cost-effective actions. We would like to highlight that a non-linear approach to trend studies is very attractive given that some air pollutants reported in this work showed not gradual-with-time reductions. Conversely, for specific pollutant source-contribution/concentration in our region, the decreasing trend was less steep at the end of the period compared to the beginning thus likely indicating the attainment of a lower limit. This was the case for example for the Secondary sulfate source contribution decreasing exponentially from 2004 to 2014 thus likely indicating a limited scope for further reduction of SO2 emissions in our region."*

**Minor/technical comments**

5) *Various sections should be called sections, not paragraphs in the text.*

      *Paragraph was replaced with Section throughout the manuscript.*

6) *Ammonium sulfate and ammonium nitrate sound a bit strange names for sources, as they are compounds that originate from a number of sources as a result of atmospheric processing.*

      The names *Ammonium sulfate* and *Ammonium nit*rate were replaced with *Secondary sulfate* and *Secondary nitrate* throughout the manuscript.

7) *Line 156: please specify EUSAAR protocol.*

*Details on the EUSAAR Protocol can be found in* Cavalli, F., Viana, M., Yttri, K. E., Genberg, J., and Putaud, J.-P.: Toward a standardised thermal-optical protocol for measuring atmospheric organic and elemental carbon: the EUSAAR protocol, Atmos. Meas. Tech., 3, 79-89, doi:10.5194/amt-3-79-2010, 2010.

This reference was added to the Bibliography.

8) *The language of the paper requires some improvements here and there. Some examples:- there is something wrong the sentence on lines 76-78 - line 133: : : :will be discussed later - line 250: : : :(not shown), we concluded: : : - line 270: : : :what was observed: : : - there is something wrong the sentence on lines 280-284.*

The language of the manuscript was revised

[revised manuscript text omitted]

BCN measuring site from 2009

Diagonal Av.

BCN measuring site before 2009

Unpaved parking   works

Barcelona Metrop. Area (BMA)

BCN

SPAIN

MSY BCN

MSY measuring site

MSY

BMA

**Figure 1**

[Figure]

**Figure 2**

[Figure]

**Figure 3**

[Figure]

**Figure 4**

[Figure]

**Figure 5**

[Figure]

**Figure 6**

[Figure]

**Figure 7**

[Figure]

**Figure 8**

[Figure]

**Figure 9**

[Figure]

**Figure 10**

**Supporting Information**

**1) *Effect of the change of the location of the measuring station in BCN in 2009**

[Figure]

**Figure SI-1**: Trends of PM$_{10}$ concentrations from gravimetric measurements at BCN and MSY. Red rectangle highlights the decrease of PM$_{10}$ concentration at BCN due to the change of the location of the BCN measuring station in 2009.

**2) *PMF source profiles at BCN and MSY**

[Figure]

**Figure SI-2**: Chemical profiles of the PMF sources at MSY (a) and BCN (b)

Source profiles from PMF analysis at BCN and MSY. Common sources at both BCN and MSY were: *Secondary Sulfate* (secondary inorganic source traced by $SO_4^{2-}$, $NH_4^+$ with contribution from OC), *Secondary nitrate* (secondary inorganic source traced by $NO_3^-$ and $NH_4^+$), *V-Ni bearing* (traced mainly by V, Ni and $SO_4^{2-}$ it represents the direct emissions from heavy oil combustion), *mineral* (traced by typical crustal elements such as Al, Ca, Ti, Rb, Sr), *aged marine* (traced by Na and Cl mainly with contributions from $SO_4^{2-}$ and $NO_3^-$). Non common sources at MSY were: *Industrial/Traffic source* (Anthropogenic source traced by EC, OC, Cr, Cu, Zn, As, Cd, Sn, Sb and Pb includes contributions from anthropogenic sources such as traffic and metallurgic) and *Aged organics* (traced by OC and EC mainly with maxima in summer indicating a biogenic origin). Non common sources at BCN were: *Traffic* (traced by Cnm, Cr, Cu, Sb and Fe mainly and contributing 5.14 µg/m³ (15.1%)), *Road resuspension* (traced by both crustal elements, mainly Ca, and traffic tracers such as Sb, Cu and Sn and contributing 4.25 µg/m³ (12.5%)) and *Industrial* (traced by Pb, Cd, As and Zn and contributing 0.96 µg/m³ (2.8%).

3) *OC:EC ratio statistic at Montseny (MSY)*

[Figure]

**Figure SI-3**: OC and EC scatterplot (a) and frequency distribution of the OC:EC ratio (b) at Montseny (MSY) station.

Mean and median values of the OC:EC ratio at MSY were 9.1 and 7.8, respectively.

**4) PMF Barcelona: 2007-2014 (with OC and EC) vs. 2004 – 2014 (with Cnm)**

[Figure]

**Figure SI-4**: Comparison between PMF results at BCN obtained using the period 2007-2014 (separate OC and EC measurements available) and using the whole period 2004-2014 (Cnm was used in PMF).

**5) OC:EC ratio statistic at Barcelona (BCN)**

[Figure]

**Figure SI-5**: OC and EC scatterplot (a) and frequency distribution of the OC:EC ratio (b) at Barcelona (BCN) station.

---

## Author Response (AR2)

**Answer to the comments from Co-Editor**

**Comments to the Author:**

*The authors have done a good job in revising this paper. I noticed a few language issues when reading the text. I recommend that before uploading the final text file, the authors go through the text once more to correct the grammar.*

As suggested by the Co-Editor the paper was checked in order to correct the grammar. Yellow colour highlights changes made in this new version of the manuscript.

[revised manuscript text omitted]